# Hemodynamic-mediated endocardial signaling controls in vivo myocardial reprogramming

**Manuel Gálvez-Santisteban[1†], Danni Chen[1†], Ruilin Zhang[2†], Ricardo Serrano[3], Cathleen Nguyen[3], Long Zhao[4], Laura Nerb[1], Evan M Masutani[1], Julien Vermot[5], Charles Geoffrey Burns[4], Caroline E Burns[4], Juan C del Álamo[3,6], Neil C Chi[1,6,7]\***

[1]Department of Medicine, Division of Cardiology, University of California, San Diego, La Jolla, United States; [2]State Key Laboratory of Genetic Engineering, School of Life Sciences, Fudan University, Shanghai, China; [3]Mechanical and Aerospace Engineering Department, University of California, San Diego, La Jolla, United States; [4]Department of Medicine, Cardiovascular Research Center, Massachusetts General Hospital and Harvard Medical School, Charlestown, United States; [5]Institut de Génétique et de Biologie Moléculaire et Cellulaire, Centre National de la Recherche Scientifique, UMR7104, INSERM U964, Université de Strasbourg, Illkirch, France; [6]Institute for Engineering in Medicine, University of California, San Diego, La Jolla, United States; [7]Institute of Genomic Medicine, University of California, San Diego, La Jolla, United States

\*For correspondence:
nchi@ucsd.edu

[†]These authors contributed equally to this work

**Competing interests:** The authors declare that no competing interests exist.

**Abstract** Lower vertebrate and neonatal mammalian hearts exhibit the remarkable capacity to regenerate through the reprogramming of pre-existing cardiomyocytes. However, how cardiac injury initiates signaling pathways controlling this regenerative reprogramming remains to be defined. Here, we utilize in vivo biophysical and genetic fate mapping zebrafish studies to reveal that altered hemodynamic forces due to cardiac injury activate a sequential endocardial-myocardial signaling cascade to direct cardiomyocyte reprogramming and heart regeneration. Specifically, these altered forces are sensed by the endocardium through the mechanosensitive channel Trpv4 to control Klf2a transcription factor expression. Consequently, Klf2a then activates endocardial Notch signaling which results in the non-cell autonomous initiation of myocardial Erbb2 and BMP signaling to promote cardiomyocyte reprogramming and heart regeneration. Overall, these findings not only reveal how the heart senses and adaptively responds to environmental changes due to cardiac injury, but also provide insight into how flow-mediated mechanisms may regulate cardiomyocyte reprogramming and heart regeneration.
DOI: https://doi.org/10.7554/eLife.44816.001

## Introduction

When confronted with pathological or physiological stress, tissues and organs can respond through the adaptive reprogramming of differentiated cell-types, which leads to repair in some instances and disease in others (*Jessen et al., 2015*; *Lin et al., 2017*; *Yui et al., 2018*). In the case of the heart, which experiences a vast array of environmental demands throughout life, cardiomyocytes (CM) also retain the capacity to adaptively reprogram and alter their fate in response to myocardial damage (*Bloomekatz et al., 2016*). In mammals, this reprogramming can lead to CM de-differentiation by activating cardiac developmental transcription factors to adaptively create more CMs for regenerating neonatal hearts via proliferation (*Porrello et al., 2011*). Moreover, while zebrafish ventricular

CMs can de-differentiate to produce more of these CMs during heart regeneration in a manner similar to neonatal mammalian hearts (*Jopling et al., 2010*; *Kikuchi et al., 2010*), zebrafish atrial CMs also retain sufficient plasticity to trans-differentiate in order to create new ventricular CMs for regenerating cardiac ventricles (*Zhang et al., 2013*). Although several signaling cues including Notch, BMP and Erbb2 participate in modulating heart regeneration (*D'Uva et al., 2015*; *Gemberling et al., 2015*; *Mahmoud et al., 2015*; *Münch et al., 2017*; *Polizzotti et al., 2015*; *Wu et al., 2016*; *Xiang et al., 2016*; *Zhang et al., 2013*; *Zhao et al., 2014*), how cardiac injury activates these pathways to initiate CM reprogramming and heart regeneration remains to be elucidated.

Recent studies have revealed that biomechanical forces generated by blood flow can contribute significantly to heart development through modulating Notch, BMP and Erbb2 signaling pathways (*Dietrich et al., 2014*; *Goddard et al., 2017*; *Heckel et al., 2015*; *Lee et al., 2016*; *Peshkovsky et al., 2011*; *Rasouli and Stainier, 2017*; *Samsa et al., 2015*; *Steed et al., 2016*; *Vermot et al., 2009*). During mammalian and zebrafish cardiac valve development, hemodynamic forces activate the endocardial flow-sensitive transcription factor Klf2 as well as Notch signaling to sculpt the atrio-ventricular valve (*Goddard et al., 2017*; *Heckel et al., 2015*; *Steed et al., 2016*; *Vermot et al., 2009*). Additionally, these forces regulate not only endocardial Notch signaling but also myocardial Erbb2 signaling to promote the proliferation and migration of CMs during cardiac trabeculation (*Lee et al., 2016*; *Peshkovsky et al., 2011*; *Rasouli and Stainier, 2017*; *Samsa et al., 2015*). Based on these findings, we thus further explored whether altered blood flow forces during cardiac injury may similarly activate these signaling pathways to control myocardial reprogramming and regeneration. Employing in vivo imaging and biophysical assays, we precisely monitored the dynamic intracardiac blood flow changes in the injured zebrafish heart, and furthermore, measured their biophysical impact on the cardiac chamber walls using hemodynamic-responsive transgenic reporters. Consequently, we discovered that zebrafish ventricle-injured hearts display altered oscillatory hemodynamic flow, which can be sensed by the endocardium through the mechanosensitive channel Trpv4 and transduced to activate Notch signaling. This endocardial Notch signaling can in turn non-cell autonomously initiate myocardial Erbb2 and BMP signaling to promote cardiomyocyte reprogramming and heart regeneration, whereas its inhibition through either using tissue-specific genetic-based strategies or altering blood flow impairs this reparative event. Altogether, these results reveal a Trpv4-mediated biomechanical signaling pathway that is able to sense and transduce hemodynamic alterations during cardiac injury to activate cardiomyocyte reprogramming responses that mediate cardiac regeneration.

## Results

### Endocardial Notch signaling controls myocardial reprogramming

Because recent studies have suggested that Notch signaling may be initiated early in response to cardiac injury in order to regulate heart regeneration (*Münch et al., 2017*; *Zhang et al., 2013*; *Zhao et al., 2014*), we investigated how cardiac injury may activate Notch signaling and furthermore how this signaling pathway controls CM reprogramming and heart regeneration. Using a *Tg(Tp1: eGFP)* Notch reporter line, we initially examined the activation of Notch signaling in *vmhc*:mCherry-nitroreductase (NTR) hearts that were ventricle-ablated with metronidazole (MTZ) treatment at 5 dpf. Consistent with previous studies (*Zhang et al., 2013*), MTZ treatment results initially in extensive CM death within the ventricle as detected by TUNEL staining (*Figure 1—figure supplement 1A–C*), and then a subsequent CM proliferative response in both ventricle and atrium as detected by both phospho-histone H3 studies and EdU staining (*Figure 1A–E,L–P*, *Figure 1—figure supplement 1D–F*). As a result, this CM proliferation leads to recovery of both ventricular tissue and contractile function at 96 hr post-MTZ treatment (hpt) (*Figure 1—figure supplement 1G,H*). Moreover, while differentiated atrial CMs cannot contribute to uninjured hearts at this stage (*Foglia et al., 2016*) (*Figure 1—figure supplement 2*), they do trans-differentiate into new ventricular CMs which help to recover the injured ventricle as previously reported (*Zhang et al., 2013*) (*Figure 1G,H*). In line with previous Notch signaling studies during cardiac regeneration (*Zhang et al., 2013*), we discovered elevated *Tp1*:eGFP Notch reporter signaling activity throughout these *vmhc*:mCherry-NTR ventricle-ablated hearts (*Figure 1—figure supplement 3A,B*). To further examine the dynamic

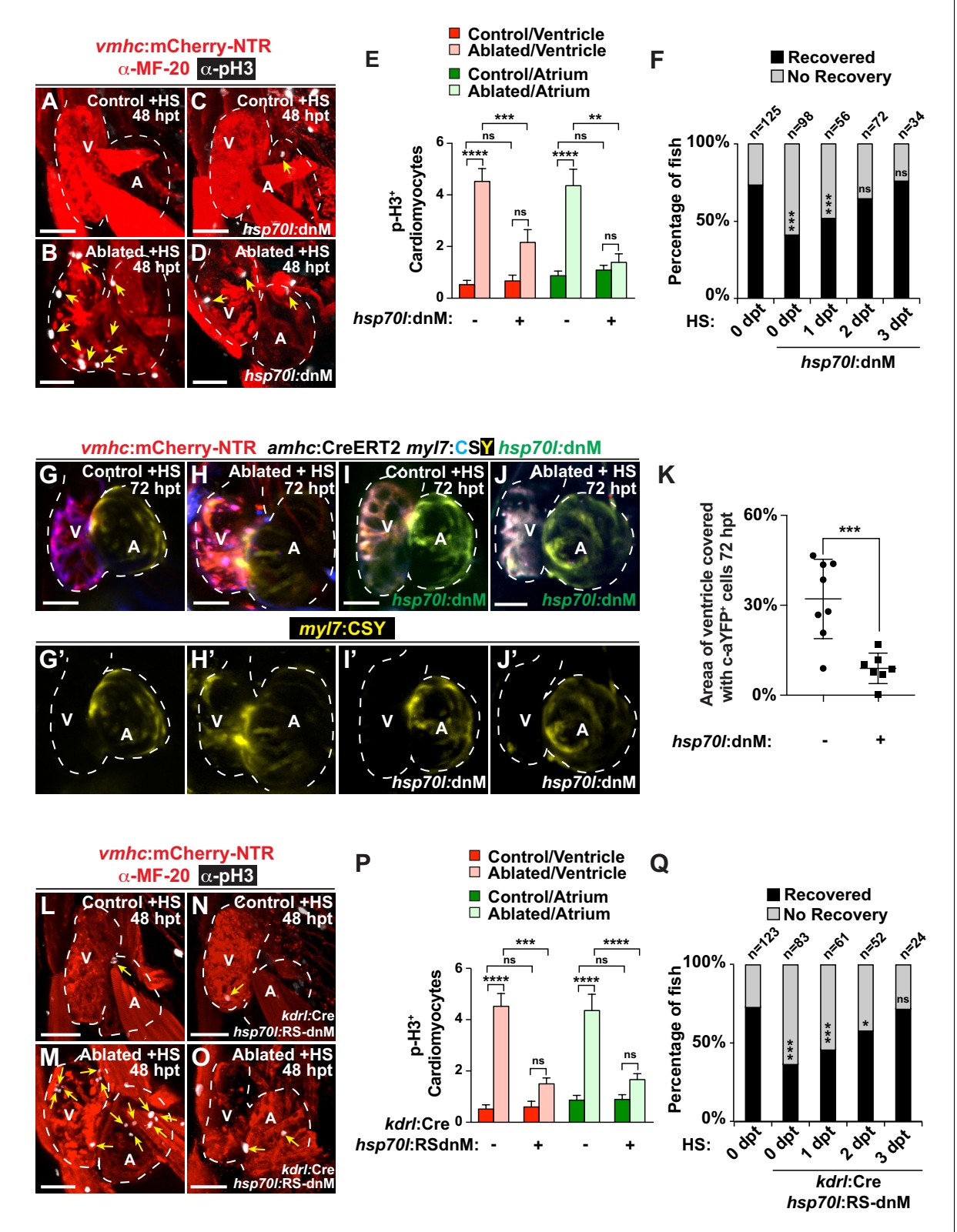

**Figure 1.** Endocardial Notch signaling controls myocardial reprogramming and cardiac regeneration. (A–D, L–O) Confocal microscopy imaging of heat-shocked/HS (A, B, L, M) *vmhc*:mCherry-NTR, (C, D) *vmhc*:mCherry-NTR; *hsp70l*:dnM and (N, O) *vmhc*:mCherry-NTR; *kdrl*:Cre; *hsp70l*:RS-dnM hearts reveals that (D) global or (O) endocardial-specific dnMAML (dnM) Notch inhibition inhibits CM proliferation in ventricle-ablated hearts at 48 hpt (7 dpf) when compared to (B, M) CM proliferation in control ventricle-ablated (no dnMAML) hearts. White – anti-phospho-histone H3 immunostaining; red –

*Figure 1 continued on next page*

*Figure 1 continued*

anti-MF-20 immunostaining. Arrows point to proliferating CMs. (E, P) Quantitation of anti-phospho-histone H3$^+$ CMs in these hearts confirms that (E) global or (P) endocardial-specific dnM Notch inhibition prevents CMs from proliferating in injured hearts (n = 23, control dnM-; 31, MTZ dnM-; 10, control dnM+; 18, MTZ dnM+; 16, control RSdnM-; 16, MTZ RSdnM-; 15, Control RSdnM+; 15, MTZ RSdnM-). Red bars – ventricle; green bars – atrium; dark bars – control sham-ablated hearts; light bars – ventricle-ablated hearts. (F, Q) Quantitation of the percentage of heat-shocked *vmhc*:mCherry-NTR, *vmhc*:mCherry-NTR; *hsp70l*:dnM and *vmhc*:mCherry-NTR; *kdrl*:Cre; *hsp70l*:RS-dnM ventricle-ablated hearts that display recovered ventricular tissue and contractility (black bars) at 96 hpt (nine dpf) shows that (F) global or (Q) endocardial-specific Notch inhibition between 0 and 1 dpt leads to the greatest inhibitory effect on overall recovery from ventricle injury. The number of fish analyzed for each condition is indicated above each column. (G–K) To examine the effects of Notch signaling on cardiac reprogramming, (G, H) *vmhc*:mCherry-NTR; *amhc*:CreERT2; *myl7*:CSY and (I, J) *vmhc*:mCherry-NTR; *amhc*:CreERT2; *myl7*:CSY; *hsp70l*: dnMAML (dnM) hearts were exposed to tamoxifen at 5 dpf to genetically label atrial CMs with YFP (c-aYFP), then (G, I) sham-ablated (control) or (H, J) ventricle-ablated, and finally heat-shocked to (I, J) induce dnM expression. Confocal microscopy imaging at 72 hpt (8 dpf) reveals that (J) heat-shock induction of dnM inhibited the ability of genetically labeled c-aYFP$^+$ atrial CMs to contribute to the regeneration of ventricle-ablated hearts when compared to (H) heat-shock control ventricle-ablated hearts. Yellow channel – (G'–J') genetically labeled c-aYFP$^+$ atrial CMs. (K) Quantitation of the percentage of ventricular area covered with c-aYFP$^+$ CMs supports that dnMAML Notch inhibition prevents atrial CMs from regenerating the injured ventricle (n = 8 *hsp70*:dnM-, seven *hsp70l*:dnM+). All confocal images shown are maximum intensity projections. V, ventricle; A, atrium; dpf, days post-fertilization; hpt, hours post MTZ/DMSO treatment. Dashed lines outline the heart. Bar: 50 µm. (E, P) Mean + s.e.m. ANOVA; (K) Mean + s.d. Student's *t*-test; (F, Q) Total numbers, Binomial test (versus 0 dpt); ns: p>0.05; *: p<0.05; **: p<0.01; ***: p<0.001; ****: p<0.0001. The following figure supplements are available for *Figure 1*.
DOI: https://doi.org/10.7554/eLife.44816.002

The following source data and figure supplements are available for figure 1:

**Source data 1.** Source data for *Figure 1* and *Figure 1—figure supplements 1–3*.
DOI: https://doi.org/10.7554/eLife.44816.007
**Figure supplement 1.** Metronidazole-ablated *vmhc*:mCherry-NTR ventricles display initial cardiomyocyte death, loss of ventricular tissue and impairment of contractile function but later exhibit cardiomyocyte proliferation and recovery of ventricular tissue and contractile function.
DOI: https://doi.org/10.7554/eLife.44816.003
**Figure supplement 2.** Reprogramming of atrial cardiomyocytes into ventricular cardiomyocytes during development.
DOI: https://doi.org/10.7554/eLife.44816.004
**Figure supplement 3.** Endocardial Notch signaling is transiently activated after myocardial injury.
DOI: https://doi.org/10.7554/eLife.44816.005
**Figure supplement 4.** Genetic inhibition of endocardial Notch signaling impairs myocardial reprogramming and regeneration.
DOI: https://doi.org/10.7554/eLife.44816.006

activity of Notch signaling, we then utilized a *Tg(Tp1:d2GFP)* Notch reporter line, which expresses a destabilized d2GFP as previously reported (*Han et al., 2016*), together with the *kdrl*:ras-mCherry endocardial/endothelial reporter line. We observed that endocardial Notch activity is specifically elevated in the endocardium within the AVC and its surrounding atrial and ventricular regions but reduced furthest away from the AVC (*Figure 1—figure supplement 3C,D*). Confirming these findings, we also detected by in situ hybridization that *notch1b* is also expressed in a similar pattern in ventricle-injured hearts (*Figure 1—figure supplement 3E,F*). Moreover, this injury-induced endocardial Notch activity peaks at 24 hpt but diminishes as the heart regenerates (*Figure 1—figure supplement 3G*). By 96 hpt, this Notch activity in the ablated hearts returns to levels similarly observed in control hearts (*Figure 1—figure supplement 3G*) and correlates to when ventricular tissue and contractility have recovered (*Figure 1—figure supplement 1G,H*).

To further interrogate the role of Notch signaling in CM reprogramming and heart regeneration, we utilized a combination of heat-shock inducible dominant negative Mastermind-like GFP (dnMAML-GFP/dnM) transgenic tools to genetically inhibit Notch signaling during cardiac injury at 5–6 dpf. Heat shock induction of *Tg(hsp70l:dnMAML-GFP); Tg(vmhc:mCherry-NTR)* ventricle ablated hearts inhibits not only the reactivation of key early cardiac transcriptional regulators such as *gata4*, *hand2* and *nkx2.5* (*Figure 1—figure supplement 4A–L*) but also overall CM proliferation in both the atrium and ventricle (*Figure 1A–E*), thus leading to reduced post-injury ventricular tissue and contractility recovery (*Figure 1F*). Through fate mapping genetically labeled atrial CMs (c-aYFP$^+$) using a combination of *Tg(amhc:CreERT2)* and *Tg(cmlc2:LoxP-AmCyan-STOP-LoxP-zsYellow)* [*Tg(myl7:CSY)*] 'switch' reporter lines, we additionally discovered that this heat-shock *hsp70l*:dnMAML-GFP Notch inhibition also prevents the ability of c-aYFP$^+$ atrial CMs to reprogram and transform into ventricle CMs (*Figure 1G–K*). Because Notch signaling is activated in the endocardium soon after cardiac injury (*Zhang et al., 2013*) (*Figure 1—figure supplement 3*), we next blocked Notch signaling

specifically in the endocardium by combining *Tg(hsp70l:LoxP-mKate2-Stop-LoxP-dnMAML-GFP)* [*Tg (hsp70l:RS-dnM)*] and *Tg(kdrl:Cre)* lines, which together allow for heat-shock induction of dnMAML-GFP specifically in the endocardium (*Figure 1—figure supplement 4M–O*), and discovered that heat-shock induced endocardial dnMAML-GFP expression (*kdrl:*Cre; *hsp70l:*RS-dnM) after *vmhc: *mCherry-NTR ventricle MTZ-ablation decreases the activation of *gata4*, *hand2* and *nkx2.5* expression (*Figure 1—figure supplement 4P–Z'*), as well as injury-induced myocardial proliferation (*Figure 1L– P*). Furthermore, time-course studies revealed that similar to global *hsp70l:*dnMAML-GFP Notch inhibition (*Figure 1F*), endocardial-specific Notch inhibition between 0 and 1 days post MTZ treatment (dpt) leads to the greatest inhibitory effect on overall recovery from ventricle injury (*Figure 1Q*), further supporting the early activation and role of endocardial Notch signaling after cardiac injury.

## Ventricular injury alters intra-cardiac hemodynamics

Because recent studies have shown that regional variations in blood flow can activate endocardial Notch signaling to regulate heart development (*Heckel et al., 2015*; *Lee et al., 2016*; *Vermot et al., 2009*), we explored whether cardiac injury could alter intracardiac blood flow and hemodynamic shear-stress to regulate injury-induced endocardial Notch signaling. Utilizing the *Tg (gata1:dsRed)* line, which labels red blood cells in dsRed, we visualized and measured in vivo blood flow dynamics by confocal microscopy and particle image velocimetry (PIV) analyses in uninjured (control) and ventricle-injured hearts at six dpf (*Figure 2—videos 1–4*). In contrast to uninjured hearts which exhibit forward blood flow from atrium to ventricle throughout the cardiac cycle (*Figure 2A,C*, *Figure 2—video 1* and *Figure 2—video 3*), we discovered that *vmhc:*mCherry-NTR ventricle-ablated hearts at 24 hpt display retrograde blood flow from the ventricle to the atrium during atrial diastole and ventricular systole (*Figure 2B,D*, *Figure 2—video 2* and *Figure 2—video 4*). Additional PIV analyses measuring blood flow velocity within various representative regions of these hearts revealed that this altered blood flow in injured hearts results in oscillatory fluctuations of positive (anterograde) and negative (retrograde) intracardiac flow (*Figure 2D–D'''*). Through calculating the fundamental harmonic index (FHI = $Q_1/Q_0$, where $Q_1$ is the amplitude of the fundamental frequency flow harmonic and $Q_0$, the time-averaged flow) (*Heckel et al., 2015*) of blood flow within the atrium, AVC and ventricle, we further quantitated and confirmed that oscillatory blood flow is significantly increased throughout these regions in ventricle-ablated hearts when compared to similar regions in uninjured control hearts (*Figure 2E*).

Given that increased levels of oscillatory flow can affect endothelial and endocardial phenotypes through modulating the expression and activity of hemodynamic responsive factors such as the transcription factor Klf2 (*Dekker et al., 2002*; *Kwon et al., 2016*; *Lee et al., 2006*), we investigated whether Klf2a expression (a zebrafish hemodynamic-sensitive paralog of Klf2) is increased in regions of the heart exhibiting increased oscillatory flow. In situ hybridizations revealed that *klf2a* expression, which primarily is located in specific regions of the AVC of uninjured hearts (*Figure 2F*), is expanded in injured hearts and present in both the AVC and the surrounding ventricular and atrial regions (*Figure 2G*). Using the *Tg(klf2a:H2B-GFP)* line, which expresses nuclear-GFP in response to changes in hemodynamic flow/stress (*Heckel et al., 2015*), we not only confirmed these findings (*Figure 2H– J*) but also created a heat-map showing that *klf2a:*H2B-GFP expression is increased in those areas of the heart exhibiting increased oscillatory flow including the AVC and its surrounding regions (*Figure 2—figure supplement 1A*). Furthermore, this *klf2a:*H2B-GFP signal overlaps with *Tp1:*d2GFP Notch reporter activity in ablated (and control) hearts (*Figure 2—figure supplement 1B*), suggesting that these signals may be modulated by blood flow dynamics. Supporting that Klf2a may regulate Notch receptor expression and/or activity in response to hemodynamic flow changes as previously suggested during valve morphogenesis in zebrafish (*Vermot et al., 2009*), we further observed that nearly all endocardial *Tp1:*nls-mCherry$^+$ cells in these hearts co-express the *klf2a:*H2B-GFP reporter (*Figure 2—figure supplement 2*). Overall, these results reveal that ventricle-injured hearts exhibit increased oscillatory flow that spatially corresponds with Klf2a expression and Notch activation.

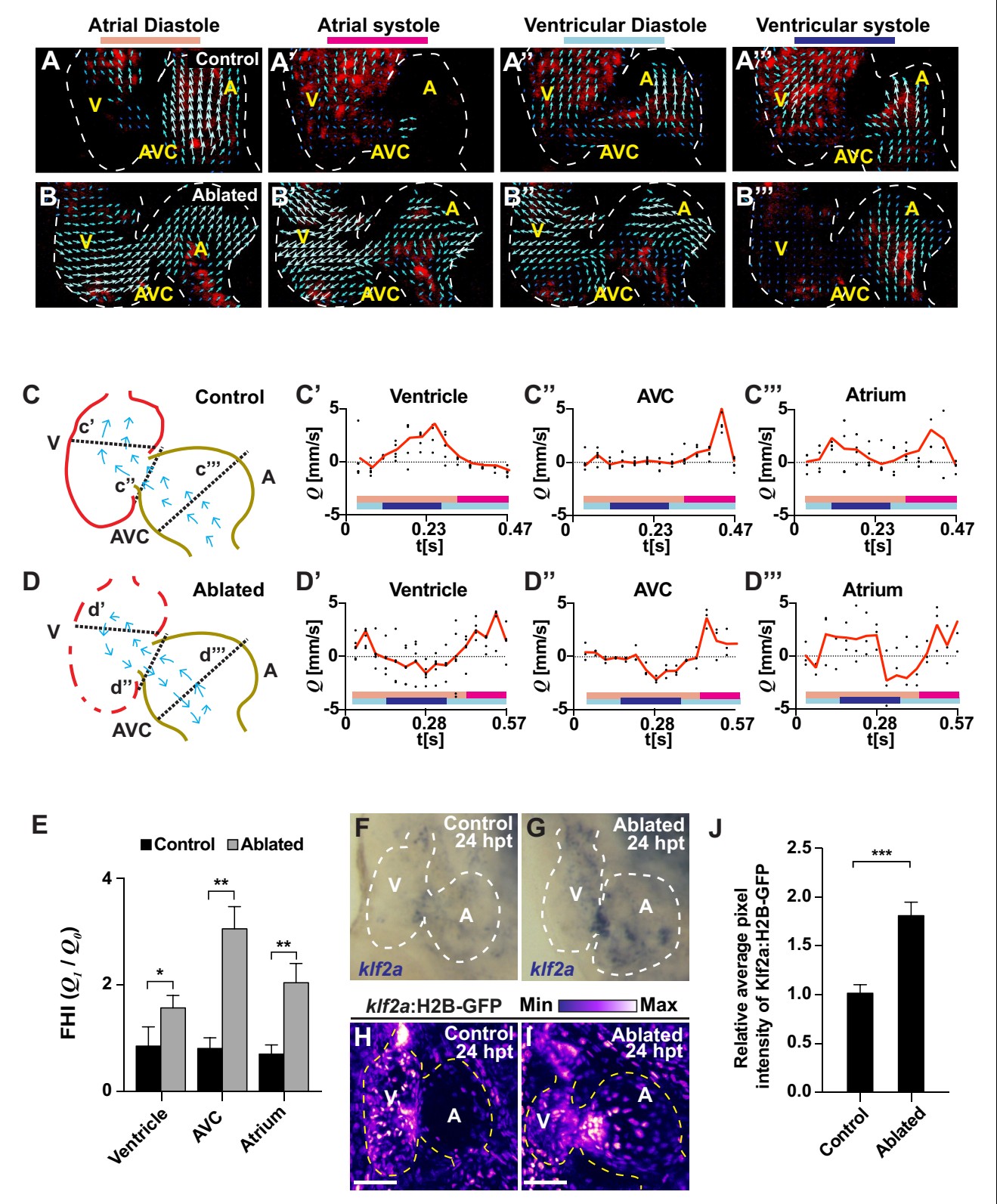

**Figure 2.** Ventricle-ablated hearts display altered oscillatory blood flow and Klf2a activation. (A, B) High-speed confocal imaging was performed on (A) *gata1*:DsRed control hearts and (B) *vmhc*:mCherry-NTR; *gata1*:DsRed ventricle-ablated hearts at 24 hpt (six dpf). Arrows represent particle image velocimetry (PIV) generated vectors from blood flow at different stages of the heart cycle: (A, B) atrial diastole, (A', B') atrial systole, (A'', B'') ventricular diastole, (A''', B''') ventricular systole. (C, D) Schematic representation indicates where flow rate (Q) was calculated in (C'– C''') control and (D'– D''')

*Figure 2 continued on next page*

*Figure 2 continued*

ventricle-ablated hearts (dotted lines). Graphs show the calculated flow rate (mm/s) in the (C', D') ventricle, (C'', D'') atrio-ventricular canal (AVC) and (C''', D''') atrium over time (t[s]). Black dots – flow rate ($Q$) in each experimental replicate. Red lines – average blood flow rate ($Q$). Experimental replicates = 5 control ventricle; 4 control AVC; 4 control atrium; 5 ablated ventricle; 3 ablated AVC; 3 ablated atrium. Colored horizontal bars on the bottom of each graph represent the cardiac cycle period: pink – atrial diastole; magenta – atrial systole; light blue – ventricular diastole; dark blue – ventricular systole. (E) Fundamental harmonic index (FHI = $Q_1/Q_0$) of the flow rate ($Q$) in the ventricle, AVC and atrium of control and ventricle-ablated hearts at 24 hpt (6 dpf). n = 3 hearts each condition. (F, G) Whole-mount in situ hybridizations show that *klf2a* expression is increased in (G) *vmhc:* mCherry-NTR ventricle-ablated hearts (n = 12/12) compared to (F) control hearts at 24 hpt (n = 0/16) (6 dpf). (H, I) Confocal imaging was performed on *klf2a*:H2B-GFP; *vmhc*:mCherry-NTR (H) control and (I) ventricle-ablated hearts at 24 hpt (6 dpf) (n = 20 each condition). (J) Quantitation of the relative average fluorescence intensity of *klf2a*:H2B-GFP in *klf2a*:H2B-GFP; *vmhc*:mCherry-NTR hearts at 24 hpt (6 dpf) confirms that *klf2a*:H2B-GFP is increased in ventricle-ablated hearts (n = 20 each condition). All confocal images shown are maximum intensity projections. V, ventricle; A, atrium; AVC, atrio-ventricular canal; dpf, days post-fertilization; hpt, hours post-MTZ/DMSO treatment. Dashed lines outline the heart. Bars: 50 µm. (E, J) Mean + s.e.m. Student's *t*-test, *, **, ***, p<0.05, p<0.01, p<0.001. The following figure supplements are available for *Figure 2*.

DOI: https://doi.org/10.7554/eLife.44816.008

The following video, source data, and figure supplements are available for figure 2:

**Source data 1.** Source data for *Figure 2* and *Figure 2—figure supplement 2*.

DOI: https://doi.org/10.7554/eLife.44816.011

**Figure supplement 1.** Klf2a and Notch are activated in areas of the heart that exhibit increased oscillatory flow.

DOI: https://doi.org/10.7554/eLife.44816.009

**Figure supplement 2.** Post-injury Notch signaling is activated in Klf2a-positive endocardial cells.

DOI: https://doi.org/10.7554/eLife.44816.010

**Figure 2—video 1.** Control hearts display anterograde intracardiac blood flow.

DOI: https://doi.org/10.7554/eLife.44816.012

**Figure 2—video 2.** Ablated hearts exhibit retrograde intracardiac blood flow.

DOI: https://doi.org/10.7554/eLife.44816.013

**Figure 2—video 2.** Particle image velocimetry (PIV) generated vectors confirm anterograde intracardiac blood flow in control hearts.

DOI: https://doi.org/10.7554/eLife.44816.014

**Figure 2—video 4.** Particle image velocimetry (PIV) generated vectors confirm retrograde intracardiac blood flow in ventricle-ablated hearts.

DOI: https://doi.org/10.7554/eLife.44816.015

## Reducing intra-cardiac hemodynamic forces impairs myocardial reprogramming

Based on these findings that Notch signaling regulates the post-injury activation of myocardial reprogramming and regeneration and furthermore is active in areas of the endocardium experiencing high oscillatory flow, we examined whether altered blood flow and hemodynamic stress in injured hearts affects myocardial reprogramming and regeneration through regulating Klf2a expression and Notch signaling. To this end, we used a combination of different genetic and chemical approaches to alter hemodynamic flow as previously reported (*Heckel et al., 2015*) to confirm findings. First, we temporally inhibited blood flow after cardiac injury by preventing cardiac contractility using blebbistatin which perturbs cardiac contractility through affecting sarcomeric function. However, because blebbistatin may also potentially impact cell proliferation and migration, we furthermore utilized tricaine, which affects electrical function, to also block cardiac contractility. Utilizing the *gata2a* mutant, which exhibits decreased blood cells (*Galloway et al., 2005*), we also lowered hemodynamic stress by reducing overall blood viscosity rather than affecting blood flow dynamics (*Boselli et al., 2017*). Under each condition, we observed that *klf2a*:H2B-GFP activity is reduced at the AVC of uninjured hearts as previously reported (*Heckel et al., 2015*) (*Figure 3E*, *Figure 3—figure supplement 1A–D*). Similarly, these conditions also reduce the activation of *klf2a*:H2B-GFP and *klf2a* expression throughout ventricle-injured hearts (*Figure 3A–I*). In line with these results, we further observed that *Tp1*:d2GFP Notch reporter activity and *notch1b* expression is also hemodynamic-sensitive in both uninjured (*Figure 3—figure supplement 1E–L*) and ventricle-ablated hearts (*Figure 3J–R*). Although Klf2a and Notch reporter and gene expression are significantly reduced after blocking contractility with blebbistatin or tricaine, we observed that they are less affected in *gata2a* mutants (*Figure 3E,N*). Based on these Klf2a and Notch findings, we further examined whether myocardial reprogramming and regeneration might also be affected under conditions of altered hemodynamic stress. For each condition, ventricle-ablated hearts display reduced activation of the early cardiac transcription factors *gata4*, *hand2* and *nkx2.5* (*Figure 4—figure supplement 1*)

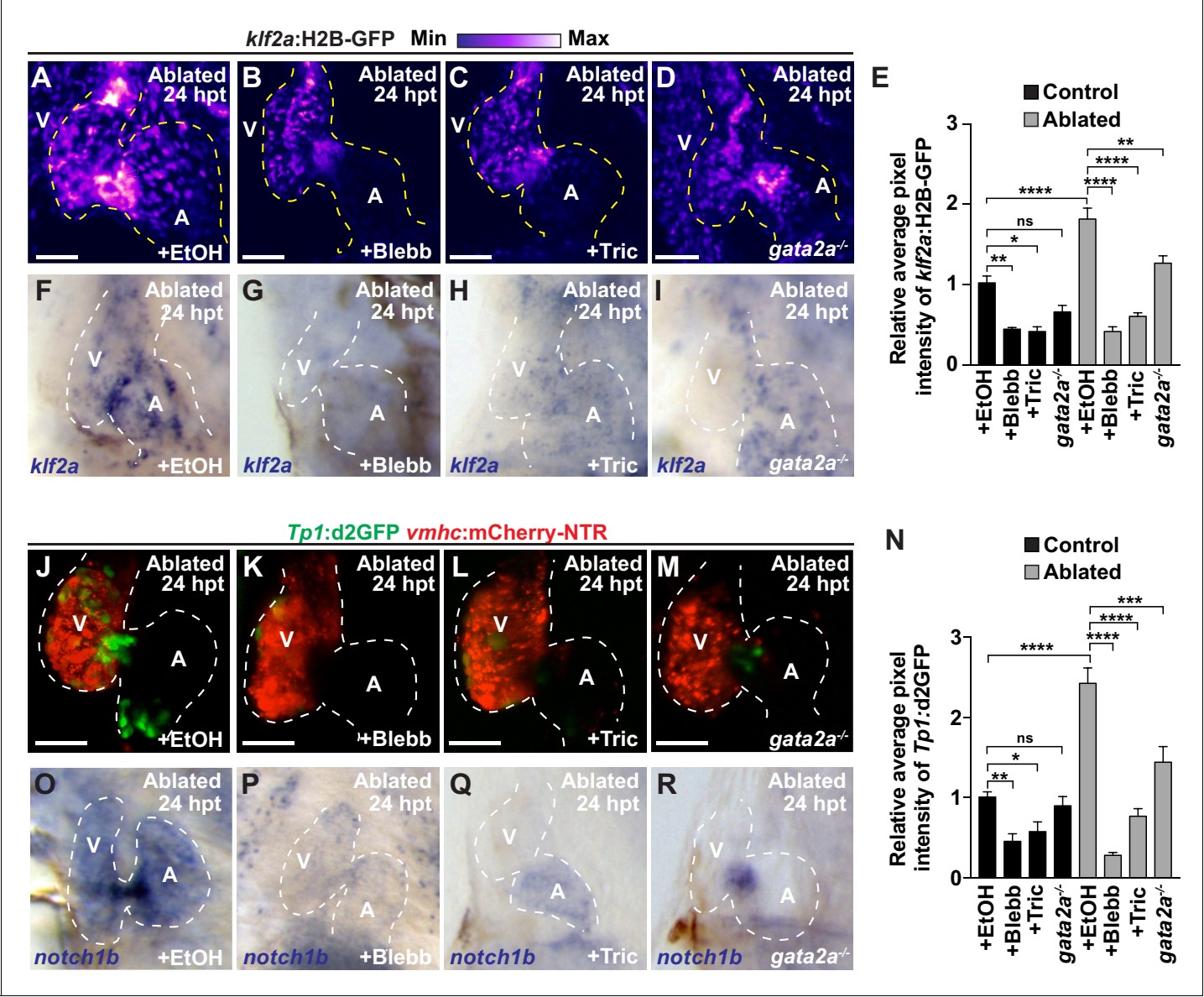

**Figure 3.** Reduced hemodynamic forces affect endocardial Notch and Klf2a post-injury activation. Confocal imaging performed on (A–D) *klf2a*:H2B-GFP; *vmhc*:mCherry-NTR or (J–M) *Tp1*:d2GFP; *vmhc*:mCherry-NTR ventricle-ablated hearts at 24 hpt (6 dpf) reveals that (D, M) *gata2a*[-/-] mutant hearts as well as wild-type hearts treated with (B, K) blebbistatin (Blebb) or (C, L) tricaine (Tric) exhibit reduced *klf2a*:H2B-GFP or *Tp1*:d2GFP expression when compared to (A, J) ethanol (EtOH) sham-treated hearts. Quantitation of the relative average fluorescence intensity in control (black bars) and ventricle-ablated (gray bars) hearts confirms reduced (E) *klf2a*:H2B-GFP or (N) *Tp1*:d2GFP expression in *gata2a*[-/-] mutant hearts as well as wild-type hearts treated with blebbistatin or tricaine (*klf2a*:H2B-GFP n = 12 control EtOh; 5 control Blebb; 7 control Tric; 9 control *gata2a*[-/-]; 11 MTZ EtOH; 6 MTZ Blebb; 6 MTZ Tric; 9 MTZ *gata2a*[-/-]. *Tp1*:d2GFP n = 12 control EtOh; 6 control Blebb; 7 control Tric; 9 control *gata2a*[-/-]; 11 MTZ EtOH; 6 MTZ Blebb; 6 MTZ Tric; 9 MTZ *gata2a*[-/-]). Whole-mount in situ hybridizations at 24 hpt (6 dpf) show that (F–I) *klf2a* and (O–R) *notch1b* expression is decreased in *vmhc*:mCherry-NTR ventricle-ablated hearts treated with (G, P) blebbistatin (n = 0/8 *klf2a*; n = 2/12 *notch1b*) or (H, Q) tricaine (n = 0/7 *klf2a*; n = 0/10 *notch1b*) as well as in (I, R) *gata2a*[-/-] ventricle-ablated hearts (n = 3/9 *klf2a*; n = 5/12 *notch1b*) when compared to (F, O) hearts sham-treated with ethanol (n = 7/7 *klf2a*; n = 10/10 *notch1b*). All confocal images shown are maximum intensity projections. V, ventricle; A, atrium; dpf, days post-fertilization; hpt, hours post-MTZ treatment. Dashed lines outline the heart. Bars: 50 µm. (E, N) Mean + s.e.m. ANOVA, ns: p>0.05; *: p<0.05; **: p<0.01; ***: p<0.001; ****: p<0.0001. The following figure supplements are available for *Figure 3*.

DOI: https://doi.org/10.7554/eLife.44816.016

The following source data and figure supplement are available for figure 3:

**Source data 1.** Source data for *Figure 3*.
DOI: https://doi.org/10.7554/eLife.44816.018

**Figure supplement 1.** Inhibiting hemodynamic flow leads to reduced cardiac Klf2a and Notch signaling.

*Figure 3 continued on next page*

*Figure 3 continued*

DOI: https://doi.org/10.7554/eLife.44816.017

as well as decreased CM proliferation (*Figure 4A–E*), which significantly impair overall recovery from ventricular injury (*Figure 4F*). Using a combination of *Tg(amhc:CreERT2)* and *Tg(β-actin2:loxP-DsRed-STOP-loxP-eGFP)* [or *Tg(β-actin2:RSG)*] lines to genetically label atrial CMs (c-aGFP+), we also discovered that reducing blood flow and viscosity under these conditions significantly reduces the contribution of c-aGFP+ labeled atrial CMs to the ventricle after ventricle ablation (*Figure 4G–K*). Altogether, these results support the role for hemodynamic forces in controlling post-injury activation of endocardial Notch and subsequent myocardial reprograming and regeneration during cardiac repair.

## Trpv4 mechanosensation of intra-cardiac blood flow regulates myocardial reprogramming

Because recent studies have shown that the mechanosensitive channel Trpv4 can sense intracardiac hemodynamic oscillatory flow and activate *klf2a* expression during cardiac valvulogenesis (*Heckel et al., 2015*), we utilized *trpv4* and *klf2a* mutants to investigate whether this biomechanical signaling pathway may also sense hemodynamic changes in the heart after ventricle injury to control Notch signaling as well as myocardial reprogramming and regeneration. Consistent with recent studies suggesting that Trpv4 controls *klf2a* expression (*Heckel et al., 2015*), we observed that *trpv4* mutants, which do not exhibit retrograde blood flow (*Figure 5—videos 1*, *2*), display reduced *klf2a*:H2B-GFP expression in not only uninjured (control) but also ventricle-ablated hearts (*Figure 5A–E*). Given that both *trpv4* and *klf2a* mutants also display reduced *Tp1*:d2GFP Notch reporter and *notch1b* expression in the uninjured (control) and ventricle-ablated hearts (*Figure 5F–L*, *Figure 5—figure supplement 1*), we further examined the ability of these mutants to regenerate ablated ventricles. Similar to reducing hemodynamic blood flow and shear stress, we discovered that *klf2a* and *trpv4* ventricle-injured mutant hearts exhibit reduced activation of early cardiac transcription factors (*Figure 5—figure supplement 2*) as well as diminished cardiomyocyte proliferation (*Figure 5M–P*), which leads to overall impaired recovery from ventricular injury (*Figure 5Q*). Corroborating these findings, lineage tracing studies further reveal that the reprogramming of genetically labeled c-aGFP+ atrial CMs into ventricular CMs to regenerate the ventricle is significantly reduced in *klf2a* and *trpv4* mutant hearts when compared to wild-type control hearts (*Figure 5R–U*). Thus, these results support the role of the mechanosensitive channel Trpv4 in regulating post-injury Notch signaling activation as well as myocardial reprogramming and regeneration through the Klf2a transcription factor.

## Hemodynamic-mediated endocardial signaling pathways activate BMP and Erbb2 signaling to regulate myocardial reprogramming and regeneration

Since hemodynamic-mediated endocardial signaling pathways can non-cell autonomously initiate BMP and Erbb2 myocardial signaling pathways during cardiac trabeculation (*Dietrich et al., 2014*; *Rasouli and Stainier, 2017*; *Samsa et al., 2015*), we investigated whether injury-induced endocardial signaling similarly activates these myocardial signaling pathways to control CM reprogramming and heart regeneration during cardiac injury. To this end, we initially examined whether BMP and/or Erbb2 signaling regulates both cardiac regeneration and myocardial reprogramming. Consistent with previous cardiac regeneration studies (*D'Uva et al., 2015*; *Gemberling et al., 2015*; *Mahmoud et al., 2015*; *Polizzotti et al., 2015*; *Wu et al., 2016*; *Xiang et al., 2016*), we discovered that blocking either BMP signaling with Dorsomorphin (DM) or Erbb2 signaling with AG1478 as previously described (*Han et al., 2016*), diminishes not only ventricle injury-induced CM proliferation (*Figure 6A–C,E*) and activation of early cardiac transcription factors (*Figure 6—figure supplement 1*), but also the trans-differentiation of genetically labeled c-aGFP+ atrial CMs into ventricular CMs (*Figure 6F–H,J*) when compared to DMSO controls. Confirming these results, we furthermore observed that *erbb2*−/− mutants exhibit similar injury-response defects compared to those observed in AG1478-treated hearts (*Figure 6D,E,I,J*). As a result, this inhibition of CM proliferation and

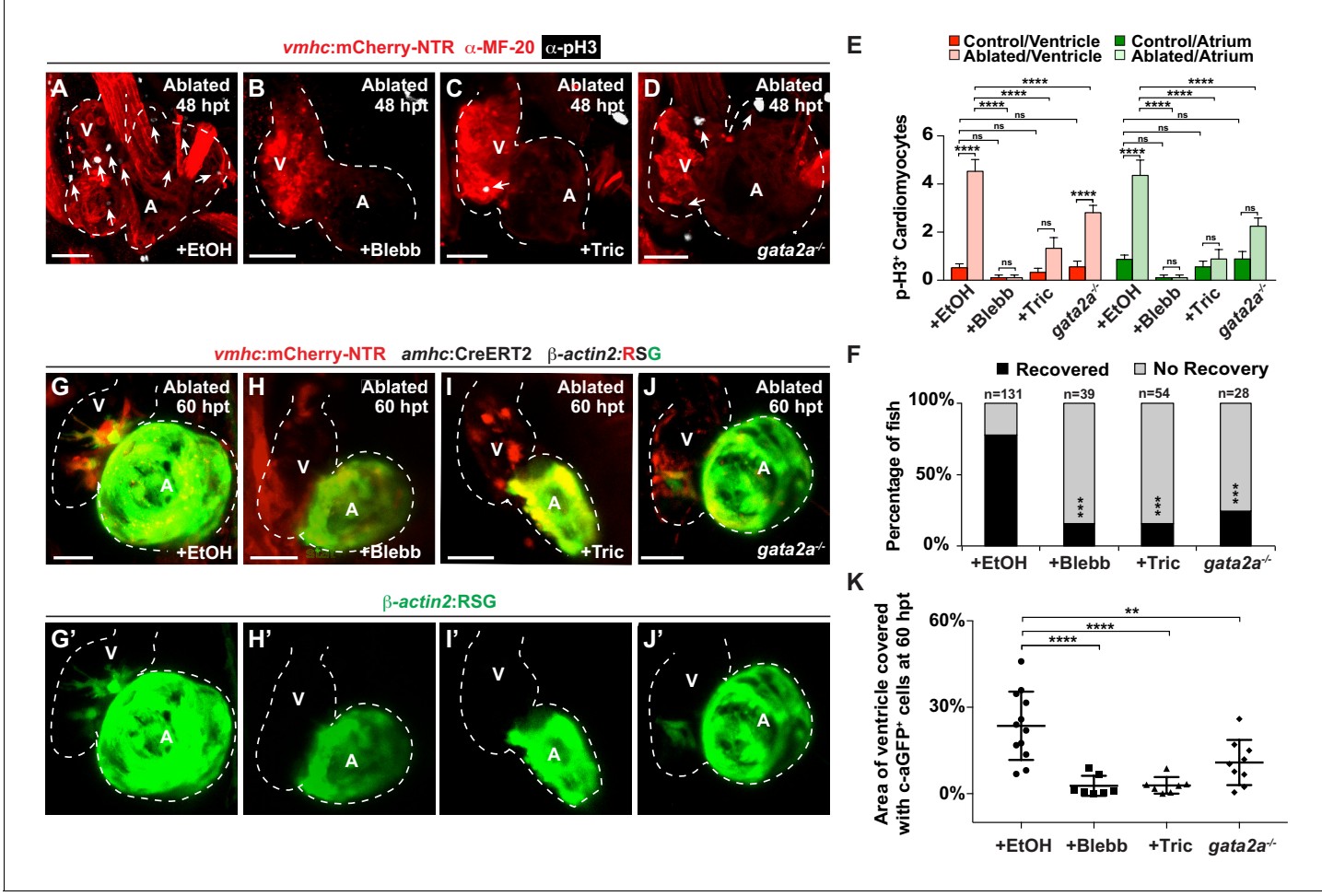

**Figure 4.** Hemodynamic forces control regeneration and myocardial reprogramming. (**A–D**) Confocal microscopy imaging performed on *vmhc:*mCherry-NTR ventricle-ablated hearts reveals that (**B**) blebbistatin or (**C**) tricaine treatment as well as (**D**) the *gata2a*[-/-] mutant allele inhibit CM proliferation at 48 hpt (7 dpf) when compared to CM proliferation of (**A**) ethanol-treated control hearts. White – anti-phospho-histone H3 immunostaining; red – anti-MF-20 immunostaining. Arrows point to proliferating CMs. (**E**) Quantitation of anti-phospho-histone H3[+] CMs in these hearts confirms that blebbistatin or tricaine treatment, and *gata2a*[-/-] defects block CM proliferation in injured hearts (n = 23, control EtOH; 21 MTZ EtOH; 9 control Blebb; 9 MTZ Blebb; 10 control Tric; 10 MTZ Tric; 9 control *gata2a*[-/-]; 16 MTZ *gata2a*[-/-]). Red bars – ventricle; green bars – atrium; dark bars – control sham-ablated hearts; light bars – ventricle-ablated hearts. (**F**) Quantitation of the percentage of *vmhc:*mCherry-NTR ventricle-ablated hearts that display recovered ventricular tissue and contractility (black bars) at 96 hpt (9 dpf) shows that inhibiting contractility by blebbistatin or tricaine treatment as well as decreasing blood viscosity using *gata2a*[-/-] mutants impair heart regeneration. The number of fish analyzed for each condition is indicated above each column. (**G–J**) Confocal microscopy imaging of *vmhc:*mCherry-NTR; *amhc:*CreERT2; *β-actin2:*RSG hearts at 60 hpt (7.5 dpf) reveals that (**H**) blebbistatin-treated, (**I**) tricaine-treated, or (**J**) *gata2a*[-/-] ventricle-ablated hearts exhibit reduced ability of genetically labeled atrial CMs (c-aGFP[+]) to contribute to the regenerating injured ventricle when compared to (**G**) ethanol sham-treated hearts. (**G'–J'**) Green channel – genetically-labeled c-aGFP[+] atrial CMs. (**K**) Quantitation of the percentage of ventricular area covered with c-aGFP[+] CMs confirms the reduced atrial CM contribution to the regenerating injured ventricle in *gata2a*[-/-] mutant hearts as well as wild-type hearts treated with blebbistatin or tricaine (n = 12 ethanol; 7 blebbistatin; 7 tricaine; 9 *gata2a*[-/-]). All confocal images shown are maximum intensity projections. V, ventricle; A, atrium; dpf, days post-fertilization; hpt, hours post-MTZ/DMSO treatment. Dashed lines outline the heart. Bars: 50 µm. (**E**) Mean + s.e.m. ANOVA; (**K**) Mean + s.d. ANOVA; (**F**) Total numbers, Binomial test (versus EtOH); ns: p>0.05; *: p<0.05; **: p<0.01; ***: p<0.001; ****: p<0.0001. The following figure supplements are available for *Figure 4*.

DOI: https://doi.org/10.7554/eLife.44816.019

The following source data and figure supplement are available for figure 4:

**Source data 1.** Source data for *Figure 4*.

DOI: https://doi.org/10.7554/eLife.44816.021

**Figure supplement 1.** Inhibiting hemodynamic flow perturbs post-injury re-activation of cardiac factors.

DOI: https://doi.org/10.7554/eLife.44816.020

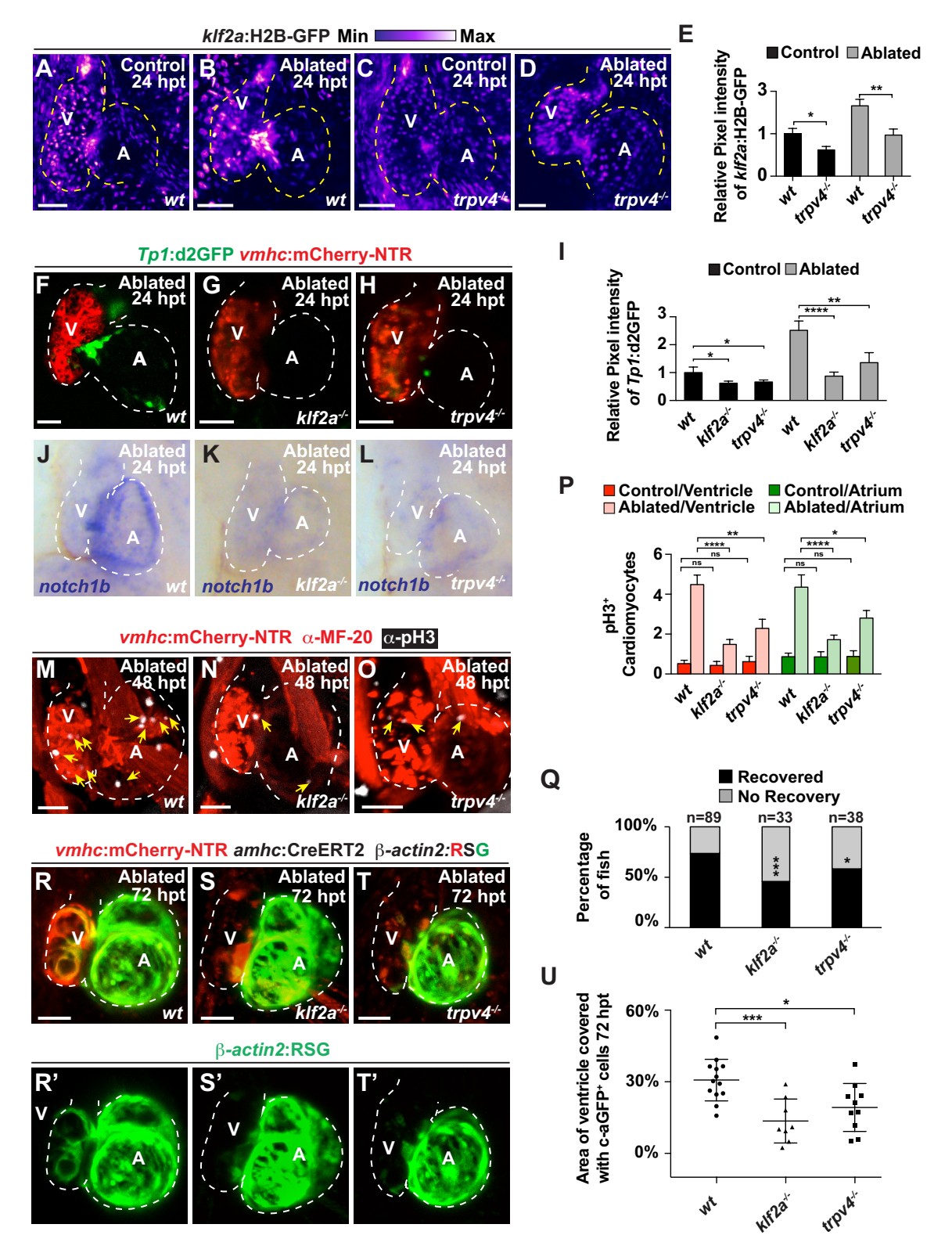

**Figure 5.** The mechanosensitive channel Trpv4 regulates endocardial Notch activation and myocardial regeneration through Klf2a. (A–D) Confocal imaging of *vmhc*:mCherry-NTR; *klf2a*:H2B-GFP hearts shows that *klf2a*:H2B-GFP expression is activated in (B) *wild-type* (*wt*) ventricle-ablated hearts at 24 hpt (6 dpf) compared to (A) control hearts; however, this activation is reduced in (D) *trpv4⁻ᐟ⁻* ventricle-ablated hearts. (F–H) Confocal imaging of *vmhc*:mCherry-NTR; *Tp1*:d2GFP hearts further reveals that *Tp1*:d2GFP is decreased in (G) *klf2a⁻ᐟ⁻* and (H) *trpv4⁻ᐟ⁻* ventricle-ablated hearts at 24 hpt (6

*Figure 5 continued on next page*

*Figure 5 continued*

dpf) when compared to (**F**) wild-type (*wt*) hearts. (**E, I**) Quantitation of the relative average fluorescence intensity confirms reduced injury-induced (**E**) *klf2a*:H2B-GFP activation in *trpv4⁻/⁻* ventricle-ablated hearts, and (**I**) *Tp1*:d2GFP activation in *trpv4⁻/⁻* and *klf2a⁻/⁻* ventricle-ablated hearts when compared to wild-type ventricle-ablated hearts (*klf2a*:H2B-GFP n = 9 control *wt*; 7 control *trpv4⁻/⁻*; 9 MTZ *wt*; 10 MTZ *trpv4⁻/⁻*. *Tp1*:d2GFP n = 15 control *wt*; 18 control *trpv4⁻/⁻*; 20 control *klf2a⁻/⁻*; 22 MTZ *wt*; 21 MTZ *trpv4⁻/⁻*; 18 MTZ *klf2a⁻/⁻*). (**J–L**) Whole-mount in situ hybridizations show that *notch1b* is decreased in ventricle-ablated (**K**) *klf2a⁻/⁻* (n = 2/11) and (**L**) *trpv4⁻/⁻* hearts (n = 6/15) at 24 hpt (6 dpf) when compared to (**J**) wild-type hearts (n = 16/18). (**M–O**) Confocal microscopy performed on *vmhc*:mCherry-NTR ventricle-ablated hearts reveals that (**N**) *klf2a⁻/⁻* and (**O**) *trpv4⁻/⁻* ventricle-ablated hearts display reduced CM proliferative response when compared to (**M**) wild-type (*wt*) ventricle-ablated hearts at 48 hpt (7 dpf). White – anti-phospho-histone H3 immunostaining; red – anti-MF-20 immunostaining. Arrows point to proliferating CMs. (**P**) Quantitation of anti-phospho-histone H3⁺ CMs in these hearts confirms that *klf2a⁻/⁻* and *trpv4⁻/⁻* hearts fail to increase CM proliferation after ventricle-injury (n = 15 each condition). Red bars – ventricle; green bars – atrium; dark bars – control sham-ablated hearts; light bars – ventricle-ablated hearts. (**Q**) Quantitation of the percentage of *vmhc*:mCherry-NTR ventricle-ablated hearts that display recovered ventricular tissue and contractility (black bars) at 96 hpt (9 dpf) supports that *klf2a⁻/⁻* and *trpv4⁻/⁻* mutants exhibit impaired heart regeneration. The number of fish analyzed for each condition is indicated above each column. (**R–T**) Confocal microscopy imaging of *vmhc*:mCherry-NTR; *amhc*:CreERT2; *β-actin2:RSG* hearts at 72 hpt (8 dpf) shows that (**S**) *klf2a⁻/⁻* and (**T**) *trpv4⁻/⁻* ventricle-ablated hearts exhibit reduced contribution of genetically labeled atrial CMs (c-aGFP⁺) to the regenerating injured ventricle when compared to (**R**) wild-type ventricle-ablated hearts. Green channel – (**R'–T'**) genetically labeled c-aGFP⁺ atrial CMs. (**U**) Quantitation of the percentage of ventricular area covered with c-aGFP⁺ CMs confirms that *klf2a⁻/⁻* or *trpv4⁻/⁻* hearts display reduced capacity to undergo injury-induced atrial-to-ventricular trans-differentiation during injury and regeneration (n = 13 *wt*; 8 *klf2a⁻/⁻*; 10 *trpv4⁻/⁻*). All confocal images shown are maximum intensity projections. V, ventricle; A, atrium; dpf, days post-fertilization; hpt, hours post-MTZ/DMSO treatment. Dashed lines outline the heart. Bars: 50 μm. (**E, I, P**) Mean + s.e.m. ANOVA; (**Q**) Total numbers, Binomial test (versus wild-type); (**U**) Mean + s.d. ANOVA; ns: p>0.05; *: p<0.05; **: p<0.01; ***: p<0.001; ****: p<0.0001. The following figure supplements are available for *Figure 5*.

DOI: https://doi.org/10.7554/eLife.44816.022

The following video, source data, and figure supplements are available for figure 5:

**Source data 1.** Source data for *Figure 5*.
DOI: https://doi.org/10.7554/eLife.44816.025
**Figure supplement 1.** *klf2a* and *trpv4* mutants display reduced endocardial Notch signaling.
DOI: https://doi.org/10.7554/eLife.44816.023
**Figure supplement 2.** *klf2a* and *trpv4* mutants show impaired post-injury re-activation of cardiogenesis transcription factors.
DOI: https://doi.org/10.7554/eLife.44816.024
**Figure 5—video 1.** High-speed bright-field imaging performed on wild-type control heart at 2 dpf shows normal blood flow.
DOI: https://doi.org/10.7554/eLife.44816.026
**Figure 5—video 2.** High speed bright-field imaging performed on *trpv4 -/-* mutant heart at 2 dpf shows normal blood flow.
DOI: https://doi.org/10.7554/eLife.44816.027

reprogramming results in the failure to regenerate the injured heart and recover from ventricle ablation (**Figure 6K**). To further investigate whether injury-induced endocardial signaling pathways regulate this myocardial BMP and/or Erbb2 signaling activation, we examined components of BMP and Erbb2 signaling pathways in ventricle-ablated hearts after reducing intracardiac hemodynamic flow with blebbistatin or inhibiting Notch signaling with either DAPT or dnMAML-GFP. Using the *Tg(BRE: d2GFP)* BMP reporter line, which has been previously used to detect BMP signaling in the myocardium of zebrafish hearts (**Han et al., 2016**), we observed that blocking either hemodynamic flow or Notch signaling decreases the activation of myocardial BMP signaling that occurs after ventricle ablation (**Figure 7A–F**, **Figure 7—figure supplement 1**). Furthermore, we examined in injured hearts the expression of *bmp10* and *nrg1*, which are developmentally expressed in the myocardium and endocardium in order to respectively activate BMP and Erbb2 signaling during cardiac trabeculation (**Grego-Bessa et al., 2007**; **Han et al., 2016**; **Rasouli and Stainier, 2017**; **Samsa et al., 2015**). We found by in situ hybridization analyses that while the expression of these factors are elevated in the heart after ventricle injury, this increased expression is blocked after inhibiting hemodynamic flow and Notch signaling (**Figure 7G–R**, **Figure 7—figure supplement 2**), suggesting that these ligands may also regulate BMP and Erbb2 during CM reprogramming and cardiac regeneration. Altogether, these findings support that hemodynamic-mediated endocardial signaling pathways may regulate CM reprogramming and heart regeneration through modulating myocardial BMP and Erbb2 signaling.

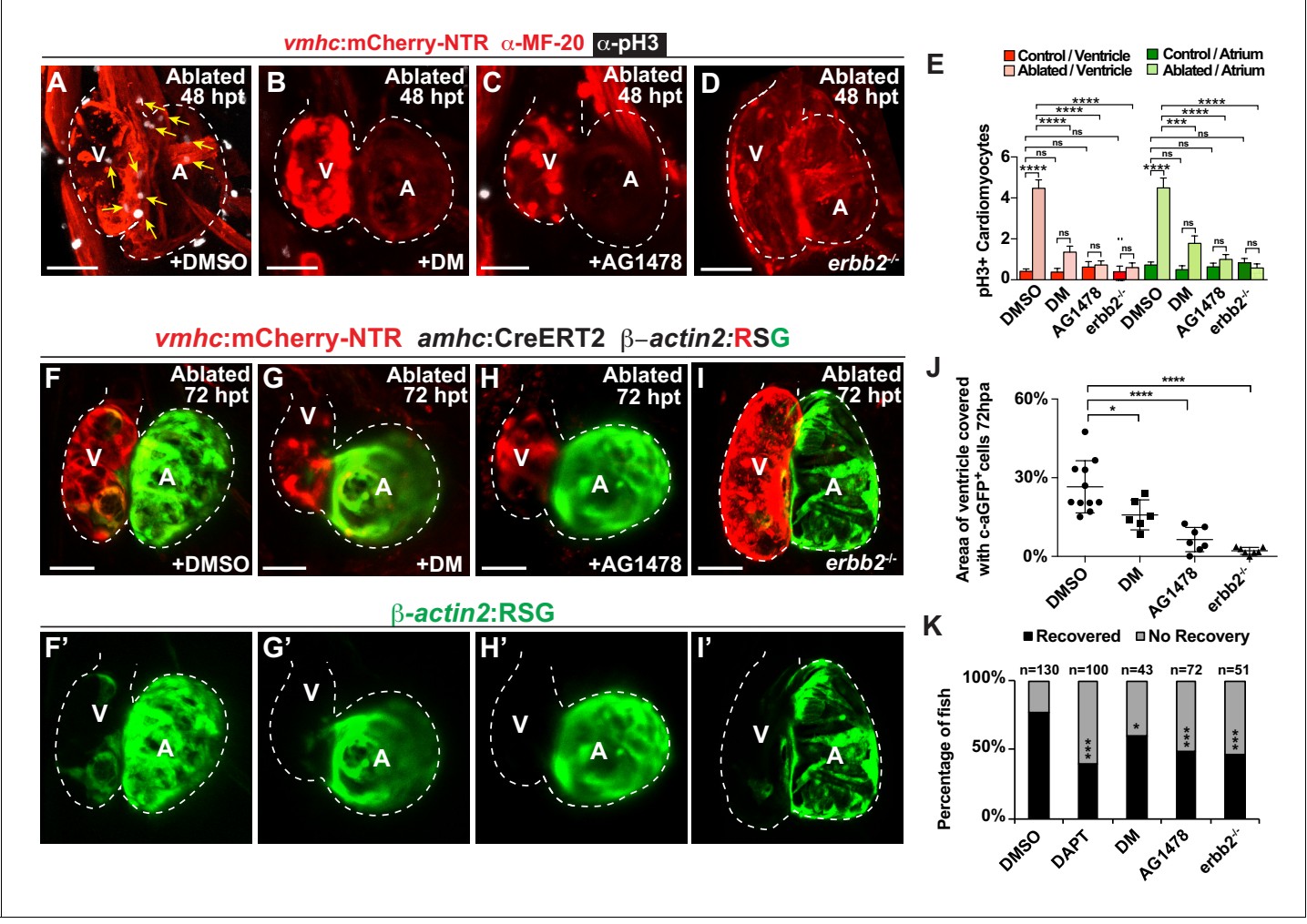

**Figure 6.** Erbb2 and BMP signaling regulate cardiac regeneration and atrial-to-ventricular trans-differentiation. (A–D) Confocal microscopy performed on *vmhc*:mCherry-NTR ventricle-ablated hearts reveals that (B) dorsomorphin (DM) and (C) AG1478-treated as well as (D) *erbb2* loss-of-function mutation (*erbb2^-/-^*) ventricle-ablated hearts exhibit reduced CM proliferative response when compared to (A) DMSO-treated ventricle-ablated hearts (control) at 48 hpt (7 dpf). White – anti-phospho-histone H3 immunostaining; red – anti-MF-20 immunostaining. Arrows point to proliferating CMs. (E) Quantitation of anti-phospho-histone H3⁺ CMs in these hearts confirms that dorsomorphin and AG1478 treatments as well as loss of *erbb2* function (*erbb2^-/-^*) prevent CMs from proliferating in injured hearts (n = 15 each condition). Red bars – ventricle; green bars – atrium; dark bars – control sham-ablated hearts; light bars – ventricle-ablated hearts. (F–I) Confocal imaging of *vmhc*:mCherry-NTR; *amhc*:CreERT2; *β-actin2:RSG* ventricle-ablated hearts shows that (G) dorsomorphin (DM) or (H) AG1478 treatment as well as (I) *erbb2^-/-^* blocks the contribution of genetically-labeled atrial CMs (c-aGFP⁺) to the regenerating ventricle-ablated hearts when compared to (F) ventricle-ablated DMSO-treated control hearts at 72 hpt (8 dpf). Green channel – (F'–I') genetically labeled c-aGFP⁺ atrial CMs. (J) Quantitation of the percentage of ventricular area covered with c-aGFP⁺ atrial CMs confirms that dorsomorphin or AG1478 treatments as well as loss of *erbb2* function (*erbb2^-/-^*) prevent atrial CMs from regenerating the injured ventricle (n = 11 DMSO; 6 dorsomorphin; 7 AG1478; 7 *erbb2* ^-/-^). (K) Quantitation of the percentage of *vmhc*:mCherry-NTR ventricle-ablated hearts that display recovered ventricular tissue and contractility (black bars) at 96 hpt (9 dpf) confirms that inhibiting BMP signaling with dorsomorphin or Erbb2 signaling with AG1478 impairs heart regeneration. The number of fish analyzed for each condition is indicated above each column. All confocal images shown are maximum intensity projections. V, ventricle; A, atrium; dpf, days post-fertilization; hpt, hours post-MTZ/DMSO treatment. Dashed lines outline the heart. Bars: 50 μm. (E) Mean + s.e.m. ANOVA; (J) Mean + s.d. ANOVA; (K) Total numbers, Binomial test (versus DMSO); ns: p>0.05; *: p<0.05; **: p<0.01; ***: p<0.001; ****: p<0.0001. The following figure supplements are available for *Figure 6*.

DOI: https://doi.org/10.7554/eLife.44816.028

The following source data and figure supplement are available for figure 6:

**Source data 1.** Source data for *Figure 6*.
DOI: https://doi.org/10.7554/eLife.44816.030

**Figure supplement 1.** Inhibiting BMP or Erbb2 signaling impairs reactivation of early cardiogenesis transcription factors.
DOI: https://doi.org/10.7554/eLife.44816.029

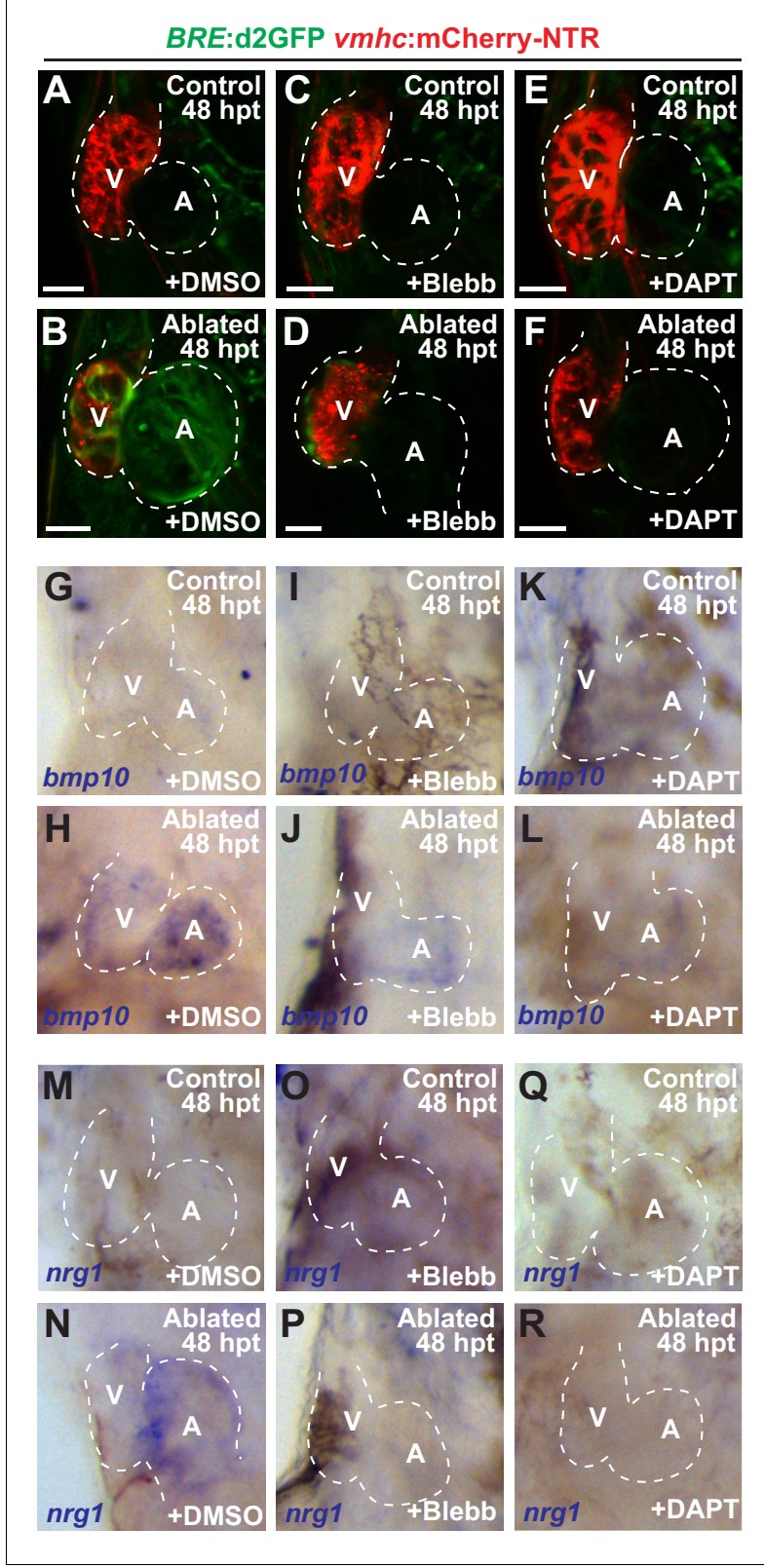

**Figure 7.** Blood flow regulates myocardial Erbb2 and BMP signaling through endocardial Notch. (**A–F**) Confocal imaging of *BRE*:d2GFP; *vmhc*:mCherry-NTR hearts at 48 hpt (7 dpf) shows that (**B**) *BRE*:d2GFP is activated after ventricular ablation (n = 21/21) when compared to (**A**) uninjured control hearts (n = 0/29); however, (**D**) blebbistatin (Blebb) (n = 0/11) or (**F**) DAPT (n = 0/14) treatment inhibits this *BRE*:d2GFP injury-induced activation. (**G–R**) Whole-
*Figure 7 continued on next page*

*Figure 7 continued*

mount in situ hybridizations reveal that *bmp10* and *nrg1* expression are increased in *vmhc:mCherry–NTR* (H, N) ventricle-ablated hearts (n = 13/14 *bmp10*; 7/8 *nrg1*) at 48 hpt (7 dpf) when compared to (G, M) control uninjured hearts (n = 0/18 *bmp10*; 0/13 *nrg1*), while treatment with (J, P) blebbistatin (n = 2/13 *bmp10*; 0/6 *nrg1*) or (L, R) DAPT (n = 4/16 *bmp10*; 0/7 *nrg1*) inhibits the injury-induced activation of *bmp10* and *nrg1*. All confocal images shown are maximum intensity projections. V, ventricle; A, atrium; dpf, days post fertilization; hpt hours post-MTZ/DMSO treatment. Dashed lines outline the heart. Bars: 50 μm. The following figure supplements are available for *Figure 7*.

DOI: https://doi.org/10.7554/eLife.44816.031

The following figure supplements are available for figure 7:

**Figure supplement 1.** Myocardial specific activation of BMP signaling.

DOI: https://doi.org/10.7554/eLife.44816.032

**Figure supplement 2.** Endocardial specific inhibition of Notch signaling blocks post-injury activation of *bmp10* and *nrg1*.

DOI: https://doi.org/10.7554/eLife.44816.033

## Discussion

Overall, these findings reveal a cardiac repair signaling mechanism that controls how the heart detects hemodynamic changes during cardiac injury in order to adaptively activate cardiomyocyte reprogramming responses mediating heart repair. These reprogramming responses include not only the re-activation of early cardiogenesis transcription factors that may revert cardiomyocytes back to a precursor state but also the conversion of atrial cardiomyocytes into ventricular cardiomyocyte to repair the injured ventricle as previously reported (*Kikuchi et al., 2010*; *Porrello et al., 2011*; *Zhang et al., 2013*). Furthermore, our results may explain why near complete ablation/injury of the entire heart (i.e. both atrial and ventricular chambers) using a *cmlc2*:NTR-based system (*Curado et al., 2007*) may result in significantly reduced heart recovery and fish survival compared to our studies as these hearts may exhibit overall weaker beating and reduced oscillatory flow compared to those where ventricular cardiomyocytes are specifically ablated but atrial cardiomyocytes are spared (*Zhang et al., 2013*). However, our findings that ventricle-cardiomyocyte ablated hearts are able to recover are in contrast to recent studies reporting that embryonic hearts develop heart failure after select ablation of a specific small population of cardiomyocytes derived from neural crest cells (*Abdul-Wajid et al., 2018*). The use of different cardiomyocyte ablation lines may potentially account for differences in outcomes between these studies as our *vmhc*:cherry-NTR lines may not be able to ablate these neural crest-derived cardiomyocytes. Alternatively, we could be ablating these neural crest-derived cardiomyocytes given that not all hearts recover from ventricular injury in our studies.

Although inflammatory and hypoxia pathways have been suggested to activate repair responses during heart regeneration (*Fang et al., 2013*; *Jopling et al., 2012*; *Nakada et al., 2017*), we discovered that the injured heart is also able to sense altered intracardiac hemodynamic forces through the endocardial biomechanical sensor Trpv4, thereby activating an endocardial-myocardial signaling pathway that directs cardiomyocyte reprogramming and heart repair. Similar to biomechanical signaling mechanisms deployed during cardiac development (*Heckel et al., 2015*), Trpv4 can transduce these forces through Klf2 to activate endocardial Notch signaling, which has been recently reported to be crucial for regulating adult zebrafish heart regeneration (*Münch et al., 2017*). In addition to its potential role in regulating inflammatory responses (*Münch et al., 2017*), we discovered that this endocardial Notch signaling may act non-cell autonomously to promote cardiac reprogramming and repair through modulating both myocardial BMP and Erbb2 signaling pathways, thus further illuminating how injury-activated endocardium may mediate myocardial regeneration as previously suggested (*Kikuchi et al., 2011*; *Münch et al., 2017*; *Zhang et al., 2013*).

Given the role of BMP and Erbb2 signaling in not only zebrafish but also mammalian adult heart regeneration (*D'Uva et al., 2015*; *Gemberling et al., 2015*; *Mahmoud et al., 2015*; *Polizzotti et al., 2015*; *Wu et al., 2016*; *Xiang et al., 2016*), our results raise the possibility that this hemodynamic-mediated cardiac repair signaling mechanism may also participate in regulating adaptive cardiac responses during mammalian cardiac injury. Furthermore, they highlight the impact of biomechanical forces in regulating adaptive cardiac regenerative responses and complement recent

mouse and zebrafish studies suggesting that cardiomyocytes and epicardial cells may sense the environment to control cardiac tissue regeneration (*Bassat et al., 2017*; *Cao et al., 2017*; *Morikawa et al., 2017*). Thus, future studies are warranted in mammalian model systems including the mouse to explore whether and how hemodynamic forces may impact mammalian heart regeneration through not only modulating BMP and Erbb2 signaling pathways but also interacting with other injury-response mechanisms including hypoxia and inflammation. As a result, such studies may reveal potential new mechanisms for how cardiac tissue may endogenously reprogram and alter their cellular differentiation state and identity to respond and adapt to environmental stresses during cardiac pathologic/disease conditions.

## Materials and methods

### Zebrafish husbandry and generation of transgenic fish lines

Zebrafish were raised under standard laboratory conditions at 28°C. All animal work we approved by the University of California at San Diego Institutional Animal Care and Use Committee (IACUC). We used the following transgenic lines: *Tg(vmhc:mCherry-NTR)$^{s957}$* (*Curado et al., 2007*); *Tg(cmlc2: GFP)$^{mss5}$* (*Dhandapany et al., 2014*); *Tg(β-actin2:loxP-DsRed-STOP-loxP-eGFP)$^{s928}$*, abbreviated *Tg (β-actin2:RSG)* (*Kikuchi et al., 2010*); *Tg(myl7:loxP-AmCyan-STOP-loxP-ZsYellow)$^{fb2}$*, abbreviated *Tg (myl7:CSY)* (*Zhou et al., 2011*); *Tg(hsp70l:dnMAML-GFP)$^{fb11}$*, abbreviated *Tg(hsp70l:dnM)* (*Zhao et al., 2014*); *Tg(cmlc2:CreERT2)$^{pd10}$* (*Kikuchi et al., 2010*); *Tg(EPV.Tp1-Mmu.Hbb: d2GFP)$^{mw43}$*, abbreviated *Tg(Tp1:d2GFP)* (*Clark et al., 2012*); *Tg(T2KTp1bglob:hmgb1-mCherry)$^{jh11}$*, abbreviated *Tg(Tp1:nls-mCherry)* (*Parsons et al., 2009*); *Tg(kdrl:ras-mCherry)$^{s896}$* (*Chi et al., 2008*); *Tg(kdrl:Cre)$^{s898}$* (*Bertrand et al., 2010*); *Tg(klf2a:H2B-GFP)* (*Heckel et al., 2015*); *Tg(gata1:DsRed)* (*Traver et al., 2003*); *and Tg(BRE-AAVmlp:eGFP)$^{mw30}$*, abbreviated *Tg(BRE:d2GFP)* (*Collery and Link, 2011*). We utilized the following lines carrying mutant alleles that were previously described and validated: *trpv4$^{sa1671}$* (*Heckel et al., 2015*), *klf2a$^{ig4}$*, (*Steed et al., 2016*), *gata2a$^{um27}$* (*Zhu et al., 2011*) and *erbb2$^{st50}$* (*Lyons et al., 2005*). The *hsp70l:loxP-mKate2-STOP-loxP-dnMAML-GFP* transgene, abbreviated as *Tg(hsp70l:RS-dnM)$^{fb24}$*, was generated as previously described (*Han et al., 2016*) with the exception that *p5E-hsp70l* (*Kwan et al., 2007*) was used as the 5' entry clone instead of *p5E-ubi*.

### Genotyping

*Trpv4$^{sa1671}$*, *klf2a$^{ig4}$* and *gata2$^{um27}$* mutants were genotyped by PCR as described (*Galloway et al., 2005*; *Heckel et al., 2015*; *Steed et al., 2016*). Adult fish, whose age is between 90 days and 2 years (https://zfin.org/zf_info/zfbook/stages/index.html), were genotyped by fin clip as described (*Westerfield, 2007*). The mutant fish utilized in each experiment were collected after imaging and subsequently genotyped as described (*Westerfield, 2007*). The *erbb2$^{st50}$* homozygous mutant embryos were identified by the previously characterized aberrant cardiac morphology (*Lyons et al., 2005*).

### Ventricular ablations

*Tg(vmhc:mCherry–NTR)* zebrafish were treated with 10 mM of Metronidazole (MTZ) (Sigma, St. Louis, MO) in egg water with 0.2% DMSO (dimethylsulphoxide) (Sigma, St. Louis, MO) for 2 hr at 28°C in the dark at 3, 4 or 5 dpf as previously described (*Zhang et al., 2013*). Control fish were simultaneously incubated in egg water with 0.2% DMSO. After the incubation, treated zebrafish were washed several times with fresh egg water and further incubated at 28°C. Efficiency of ventricular ablation and post-injury ventricular recovery was evaluated at 24 and 96 hr post MTZ/DMSO treatment (hpt), respectively. Efficiency of ventricular ablation was assessed by TUNEL cell death quantitation at 24 hpt, and by loss of ventricular tissue and contractility by imaging analyses. Efficiency of ventricular regeneration was assessed by observation of ventricular tissue and contractility by imaging analyses. Recovered fish were quantified as those fish that displayed recovered ventricular tissue and contractility at 96 hpt as previously described (*Zhang et al., 2013*). Fish that did not recover include those that displayed lack of ventricular tissue integrity and/or impaired contractility as well as fish that did not survive post-ablation.

## Quantitation of cell death

TUNEL staining was performed using the in situ cell death detection kit, fluorescein from Roche (11684795910). Briefly, 24 hpt zebrafish were fixed overnight in 4% PFA at 4°C, washed in Phosphate Buffered Saline with 0.1% Tween 20 (PBT) and dehydrated through a graded methanol series, and then incubated in 100% methonol at −20°C for at least 2 hr before graded rehydration to PBT. Samples were then incubated in 10 μg/mL proteinase K for 1 hr at room temperature, washed three times with PBT and refixed in 4% PFA for 20 min. After washing three times in PBT, samples were incubated with TUNEL staining solution overnight at 37°C and then washed five times with PBT before acquiring images using a Nikon C2 confocal microscope.

## Cardiac contractile analysis

Live zebrafish were embedded in 1% low melting agarose in a glass-bottom culture dish (MatTek), and the heart contraction was recorded by an Andor iXon EMCCD camera at a frame rate of 40 ms/frame. The fractional area change was calculated as FAC = (End diastolic area-End systolic area)/End diastolic area x 100%.

## Quantitation of ventricular area

To determine the ventricular area and the subsequent tissue recovery in ablated and DMSO control fish, ablated and control *Tg(vmhc:mCherry-NTR)* fish hearts were imaged and analyzed at 0 hpt (before ablations), 24 hpt and 96 hpt using a Nikon C2 confocal microscope (Tokyo, Japan) where an average of 25 stacks every 5 μm were acquired. To specifically determine the ventricular area, 3D reconstructions of these fish hearts were analyzed to calculate the total area covered by *vmhc*:mCherry-NTR cells using ImageJ imaging analyses (NIH, Bethesda, MD). Total ventricular area was normalized to the average 0 hpt non-ablated heart ventricular area.

## In situ hybridization

Whole-mount in situ hybridizations were performed as previously described (*Chi et al., 2008*), using the following probes: *bmp10*, *gata4*, *klf2a*, *hand2*, *nkx2.5*, *notch1b* and *nrg1*. Fish were analyzed under a Leica M205 FA stereo microscope. Magnifications of bright field images of the area containing the fish heart are shown. Blue corresponds with riboprobe staining. Heart shape was outlined based on anatomic observation under the microscope. Quantitation of the number of hearts showing increased staining of the corresponding marker as compared to non-ablated controls (numerator, n) vs total number of observed hearts is provided in each corresponding figure legend (denominator, d) (n/d).

## Heat-shock induction of dnMAML expression

Fish containing the *hsp70l*:dnM or the *hsp70l*:RS-dnM transgenes were treated with DMSO (control) or MTZ (ablated) as described. Immediately after DMSO/MTZ treatment, fish were heat-shocked for 5 min at 42°C followed by 1 hr at 37°C. Heat-shock was repeated every 12 hr until corresponding analysis was performed. Expression of dnMAML-GFP was detected under fluorescence microscopy.

## Immunofluorescence and quantification of proliferation

Immunofluorescence staining was performed as previously described for anti-phospho-histone H3 studies (*Zhang et al., 2013*). Red fluorescence corresponding to the *vmhc*:mCherry-NTR transgene was lost after fixation treatment; thus anti-MF20/anti-MHC was used to detect myocardium for these studies. The primary antibodies used in this study include: anti-GFP (chicken; Aves Labs; Tigard, OR); anti-MF20/anti-MHC (mouse; Developmental Studies Hybridoma Bank) (*Stainier and Fishman, 1992*); and anti-phospho-histone H3 (rabbit; Upstate, Lake Placid, NY). The secondary antibodies used in this study include: Alexa Fluor 594 goat anti-mouse IgG (Invitrogen, Carlsbad, CA), Alexa Fluor 633 goat anti-rabbit IgG (Invitrogen, Carlsbad, CA) and anti-chicken IgY-FITC (Sigma, St. Louis, MO). For EdU staining, we injected 2 nL EdU solution into the circulation of 48 hpt zebrafish as previously described (*Hesselson et al., 2009*). Detection of incorporated EdU was performed according to the manufacture's instructions for Click-iT 647 kit from Invitrogen (C10340). Fluorescent images were obtained using a Nikon C2 confocal microscope (Tokyo, Japan) and analyzed with ImageJ (NIH, Bethesda, MD). An average of 25 serial 5 μm z-stacks was acquired to perform 3D

reconstructions of the hearts. Proliferation was evaluated by quantification of phospho-histone H3$^+$/MF20$^+$ double positive or EdU$^+$/cmlc2:GFP$^+$ double postive cells within the ventricle and the atrium of control and ablated fish.

### Tamoxifen treatment and lineage tracing

Lineage tracing was performed as previously described (*Zhang et al., 2013*). *Tg(vmhc:mCherry–NTR; amhc:CreERT2; β-actin2:RSG)* or *Tg(vmhc:mCherry–NTR; amhc:CreERT2; myl7:CSY)* zebrafish were treated with a 10 mM 4-hydroxytamoxifen (4-OHT) solution (Sigma) or with 0.1% ethanol (unlabeled control) at 3, 4 or 5 dpf for 4 hr at 28°C and subsequently washed with fresh egg water several times. Hearts with GFP/YFP genetically labeled atrial CMs were treated with DMSO (control) or MTZ (ablated) at 12 hr post 4-OHT labeling (3.5, 4.5 or 5.5 dpf) and analyzed under a fluorescent scope every 12 hr up to 4 days. To identify ventricular GFP-positive CMs, hearts from *Tg(vmhc:mCherry–NTR; amhc:CreERT2; β-actin2:RSG)* were imaged with a Nikon C2 confocal at 60, 72 or 96 hpt as previously described (*Zhang et al., 2013*). To identify YFP-positive CMs in heat-shocked *Tg(vmhc:mCherry–NTR; amhc:CreERT2; myl7:CSY)* or *Tg(vmhc:mCherry–NTR; amhc:CreERT2; myl7:CSY; hsp70l:dnM)* hearts, zebrafish were imaged with a Nikon C2 spectral detector (Tokyo, Japan) and a 488 laser. YFP signals were obtained by extracting the minimum matching YFP wavelength from spectral data after calibration with YFP only positive controls as previously described (*Valm et al., 2016*). To determine the percentage of Cre-labeled GFP or YFP cardiomyocytes (c-aGFP/YFP$^+$) in the ventricle, maximum intensity projections of at least 25 serial 5 µm z-stacks were used to create 3D reconstructions. The ventricles were outlined based on the morphology observed in the brightfield channel. Similarly, the surface area covered by c-aGFP$^+$ or c-aYFP$^+$ CMs was outlined based on presence of GFP or YFP fluorescence. The total surface area of the outlined ventricles and the outlined c-aGFP$^+$ and c-aYFP$^+$ CMs was calculated by ImageJ (NIH, Bethesda, MD). The percentage of the ventricle covered by c-aGFP or c-aYFP was obtained by dividing the c-aGFP or c-aYFP area by the total surface area in the ventricle.

### Quantitation of transgene mean fluorescence intensity

To determine the mean fluorescence intensity, *Tg(Tp1:d2GFP)* or *Tg(klf2a:H2B:GFP)* hearts were imaged with a Nikon C2 confocal microscopy at the indicated time points. An average of 25 serial 5 µm z-stacks were acquired. Mean fluorescence levels were calculated using methods previously described (*McCloy et al., 2014*). Maximum intensity projections of confocal images were utilized to determine the region of interest (ROI), which corresponded with the zebrafish heart. Each region of interest (ROI) along with several adjacent background readings were outlined, and the area and mean fluorescence intensities were calculated by ImageJ (NIH, Bethesda, MD). The corrected fluorescence of the ROI was calculated by subtracting the background reading and normalized to the total area of the ROI. To calculate the fold increase, each ROI corrected fluorescence was divided by the corresponding control condition. Images shown are representative pictures of each condition.

### Quantitation of Tp1/Klf2a:H2B-GFP-positive cells

To determine the number of *Tp1*:nls-mCherry/*klf2a*:H2B-GFP double positive cells, *Tg(klf2a:H2B:GFP; Tp1:nls-mCherry; vmhc:mCherry-NTR)* control and ablated fish hearts were imaged with a Nikon C2 confocal microscopy at the indicated time points. An average of 25 serial 5 µm z-stacks were taken, and then images were analyzed with ImageJ (NIH, Bethesda, MD). Individual stacks were analyzed to determine the presence of *Tp1*:nls-mCherry and/or *klf2a*:H2B-GFP positive nuclei in the atria and the AVC. In the ventricle, *Tp1*:nls-mCherry positive cells were identified as those nuclei that were anatomically located beneath the myocardial layer. The myocardial layer was identified as *vmhc*:mCherry-NTR positive/*klf2a*:H2B-GFP negative.

### High-speed video acquisition

Blood flow videos were obtained by imaging control and ablated *Tg(gata1:DsRed; vmhc:mCherry-NTR)* fish at six dpf (24 hpt). Fish were imaged with a Leica Sp5 (Wetzlar, Germany) resonance scanner at 50–100 frames per second, at 512 × 128 pixels of resolution. 10 z-stacks were obtained, each separated by 10 µm for a total of 2 min. Bright field blood flow videos for two dpf wild-type control

and *trpv4 -/-* mutant were recorded by an Andor iXon EMCCD camera at a frame rate of 40 ms/frame.

## Multigrid ensemble micro particle image velocimetry

To determine the velocity of the blood inside the ventricle from obtained microscopy videos, an in-house multigrid ensemble particle image velocimetry (PIV) software was developed, as previously described (*Lindken et al., 2009*; *Adrian and Westerweel, 2011*). PIV determines deformation maps from two consecutive images in a time-lapse sequence by dividing each image into smaller interrogation windows, and maximizing the spatial cross-correlation between the matching windows of each image. In our method, the signal-to-noise ratio was improved by averaging the cross-correlation for an ensemble of window pairs coming from the same phase of the cardiac cycle across all the cycles of each video recording (500 frames, approximately 17 cycles). To calculate this ensemble average at each phase of the cardiac cycle, we temporally align frames by maximizing the temporal cross-correlation between whole-image pairs. To increase the spatial resolution, we perform multi-grid PIV (multiple PIV passes using progressively smaller interrogation windows, where each pass uses the result from the previous pass to re-center the interrogation windows). We balanced signal-to-noise ratio with resolution, starting with coarse interrogation windows of size 48 × 48 pixels and a separation of 24 pixels, and progressively refining to 16 × 16 pixel windows with a four-pixel separation, which yielded a spatial resolution of 7 microns.

## Flow rate calculations

Flow rates through different sections of the cardiac chambers were calculated from the PIV measurements using

$$Q = \int_a^b \vec{v} \cdot \vec{n} \, dl$$

where $\vec{v}$ is the velocity vector obtained by PIV, $\vec{n}$ is the vector normal to the user-drawn cross-section (i.e. lines), and $dl$ (= 4 pixels) is the discrete length spacing used to numerically calculate the integral. Multiple lines were drawn in/across the desired regions of the heart (ventricle, atrium or AVC) by selecting two points, a and b, per line and obtaining measurements of flow velocity and direction. Three to five measurements performed in at least two different stacks were used to calculate the average flow velocity and direction in each specific region. The calculated average flow velocity and direction was represented versus time for the total duration of a complete heart cycle. This plot was used to create the specific flow profiles for each region of interest.

## Fundamental harmonic index calculations

The Fundamental Harmonic Index was computed as the ratio $Q_1/Q_0$ between the amplitudes of the fundamental frequency flow harmonic (1st harmonic, $Q_1$) and the time-averaged flow (0th harmonic, $Q_0$) (*Heckel et al., 2015*). These amplitudes were determined from the first and second coefficients of the Fourier transform of the previously calculated flow profiles from ventricles, AVC and atria, using the FFT function in MATLAB (Mathworks, Natick, MA). The profiles of three different hearts were used for each condition.

## Small molecules and chemical treatments

To decrease cardiac contractility after MTZ (ablated) or DMSO (control) treatments, fish were immediately incubated in egg water with 1 mg/ml of tricaine (Sigma, St. Louis, MO) or 10 µM of blebbistatin (Sigma, St. Louis, MO) for 12 hr and then washed three times for analyses. To inhibit BMP or Erbb2 signaling after MTZ (ablated) or DMSO (control) treatments, fish were immediately treated with 10 µM Dorsomorphin (DM) (Sigma, St. Louis, MO) or 5 µM AG1478 (Sigma, St. Louis, MO), respectively. Fish were treated for 24, 48, or 72 hr as indicated. To inhibit BMP, Erbb2 or Notch signaling after MTZ (ablated) or DMSO (control) treatments, fish were immediately treated with 10 µM Dorsomorphin (DM) (Sigma, St. Louis, MO), 5 µM AG1478 (Sigma, St. Louis, MO), or 10 mM DAPT (Sigma, St. Louis, MO) until indicated time for each experiment.

## Myocardial BMP activation

To study the myocardial activation of BMP signaling, 5 dpf *Tg(BRE:d2GFP; vmhc:mCherry-NTR)* fish were treated with DMSO (control) or MTZ (ablated) as described. Fish were subsequently treated with DMSO, DAPT or blebbistatin as described above. To determine the presence of *BRE*:d2GFP, hearts were imaged with a Nikon C2 confocal microscopy at 48 hpt. An average of 25 serial 5 μm stacks was acquired for each heart. To identify *BRE*:d2GFP/*vmhc*:mCherry-NTR double positive co-expressing cells, individual stacks were analyzed with ImageJ (NIH, Bethesda, MD).

## Statistical analysis

Box plots and statistical analysis were performed using GraphPad Prism 7. ANOVA, un-paired Student's *t* or Binomial tests were used as indicated in the corresponding figure legends.

## Ethics

Animal experimentation: All procedures involving animals were in accordance with the National Institutes of Health Guide for the care and use of Laboratory Animals, and approved by the University of California at San Diego Institutional Animal Care and Use Committee (IACUC) (Protocol number S08316).

## Acknowledgements

We thank N Tedeschi for fish care and L Zhang for technical help. We thank Chi lab members for comments on the manuscript. This work was supported in part by grants from the NIH to NCC, JCDA, CGB (R01 HL127067), CEB (R01 HL127067), and by the d'Arbeloff MGH Research Scholar Award to CEB. MGS and LZ were recipients of the American Heart Association fellowship (16POST27260061 and 14POST20380738). MGS was also supported by a Fundación Ramón Areces fellowship.

## Additional information

### Funding

| Funder | Grant reference number | Author |
| --- | --- | --- |
| National Institutes of Health | | Juan C del Álamo<br>Neil C Chi |
| American Heart Association | Postdoctoral Fellowship | Manuel Gálvez-Santisteban<br>Long Zhao |
| National Institutes of Health | R01HL127067 | Charles Geoffrey Burns<br>Caroline E Burns |
| Massachusetts General Hospital | Research Scholar Award | Caroline E Burns |
| Fundación Ramón Areces | Fellowship | Manuel Gálvez-Santisteban |

The funders had no role in study design, data collection and interpretation, or the decision to submit the work for publication.

### Author contributions

Manuel Gálvez-Santisteban, Conceptualization, Resources, Data curation, Formal analysis, Validation, Investigation, Visualization, Methodology, Writing—original draft, Project administration, Writing—review and editing; Danni Chen, Formal analysis, Investigation, Visualization, Methodology, Writing—review and editing; Ruilin Zhang, Conceptualization, Supervision, Investigation, Methodology, Project administration, Writing—review and editing; Ricardo Serrano, Software, Investigation, Methodology, Writing—review and editing; Cathleen Nguyen, Software, Methodology, Writing—original draft; Long Zhao, Resources, Investigation, Methodology, Creation and validation of the hsp70l:RS-dnM zebrafish transgenic line; Laura Nerb, Formal analysis, Investigation, Visualization, Writing—review and editing; Evan M Masutani, Software, Methodology;

Julien Vermot, Resources, Supervision, Methodology; Charles Geoffrey Burns, Caroline E Burns, Resources, Supervision, Supervised the creation of the hsp70l:RS-dnM zebrafish transgenic line; Juan C del Álamo, Conceptualization, Resources, Software, Supervision, Investigation, Visualization, Methodology, Project administration, Writing—review and editing; Neil C Chi, Conceptualization, Resources, Data curation, Formal analysis, Supervision, Funding acquisition, Investigation, Visualization, Methodology, Writing—original draft, Project administration, Writing—review and editing

## Author ORCIDs

Manuel Gálvez-Santisteban (iD) https://orcid.org/0000-0003-3804-1513
Julien Vermot (iD) http://orcid.org/0000-0002-8924-732X
Charles Geoffrey Burns (iD) http://orcid.org/0000-0002-5812-6621
Neil C Chi (iD) https://orcid.org/0000-0003-2324-3796

## Ethics

Animal experimentation: This study was performed in strict accordance with the recommendations in the Guide for the Care and Use of Laboratory Animals of the National Institutes of Health. All of the animals were handled according to approved institutional animal care and use committee (IACUC) protocols (#S08316) of the University of California, San Diego.

## Decision letter and Author response

Decision letter https://doi.org/10.7554/eLife.44816.039
Author response https://doi.org/10.7554/eLife.44816.040

## Additional files

### Supplementary files

• Transparent reporting form
DOI: https://doi.org/10.7554/eLife.44816.034

### Data availability

All data generated or analysed during this study are included in the manuscript and supporting files.

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
