## [Decision Letter]

Thank you for sending your article entitled "Hemodynamic-mediated endocardial signaling controls *in vivo* myocardial reprogramming" for peer review at *eLife*. Your article is being evaluated by Marianne Bronner as the Senior Editor, a Reviewing Editor, and three reviewers.

All of the reviewers see merit in the manuscript but also raise issues that will require extensive revisions. Our concern is that these revisions may take a considerable time beyond the two to three months required for a revision in *eLife*. Given the list of essential revisions, including new experiments, the editors and reviewers invite you to respond within the next two weeks with an action plan and timetable for the completion of the additional work. We plan to share your responses with the reviewers and then issue a binding recommendation. The reviewers' comments are detailed below.

*Reviewer #1:*

In this manuscript, Chi and colleagues describe an endocardial to myocardial signalling pathway that mediates cardiomyocyte (CM) reprogramming and proliferation in the embryonic zebrafish ventricular ablation model. Increased oscillatory flow, sensed in part by the Trpv4 mechanosensory channel, is required to activate endocardial Klf2 and Notch activity, which in turn influences Bmp and Erbb2 activity in CMs. While these components have all been shown to influence cardiac development, and in many cases adult cardiac regeneration, this work provides a holistic analysis of this pathway in the embryonic setting.

Overall, the manuscript is clearly written, with well presented and supported experimental results.

Comments:

1) The data for the Notch response in Figure 1—figure supplement 2 is somewhat confusing. Heightened Notch reporter activity in the ventricle is shown/discussed (panels A and B), yet Notch1b expression in panel F is clearly atrial, yet called "throughout the heart" in the text. This should be clarified.

2) In Figure 1—figure supplement 3 panel L, *nkx2.5* appears to be induced in the atrium, but not ventricle, following Notch inhibition. Does this imply incomplete reprogramming of atrial CMs? Again, this should be discussed. Are these representative images?

3) In Figure 2—figure supplement 1, it is unclear what is being counted as a *klf2/tp1* (Notch) double-positive cell, especially given the red colour of the NTR transgene as well.

4) The authors present and support a model where alterations/increases in oscillatory flow trigger an endocardial to myocardial cascade that initiates cardiac reprogramming/proliferation. However, regeneration of the embryonic heart following full (*cmlc2*:NTR-based) CM ablation has been reported (Curado, 2007/2008), where presumable blood flow is minimal.

*Reviewer #2:*

In this manuscript, Gálvez-Santisteban et al., use a nice set of biophysical and genetic assays to characterize injury-induced cellular responses in embryonic hearts. The main conclusion is that cardiac injury alters oscillatory blood flow and induces Trpv4-mediated Klf2-Notch signaling in endocardial cells to regulate cardiomyocyte reprogramming and regeneration in *erbb2* and *bmp* dependent manner. While this manuscript is clearly written and the data are nicely presented, a better characterization of the injury model and more robust assays for regeneration are needed before publication.

1) In this study, the assessment of cardiac regeneration is primarily based on pH3 staining (~4 cells per heart). More robust assays for cardiomyocyte proliferation and cardiac function will strengthen the data.

2) A recent study shows that the embryonic heart cannot fully recover from the ablation of a small population of cardiomyocytes. The injured hearts develop trabeculation defects which ultimately lead to heart failure. In this study, the ablation covers the entire ventricle, yet the majority of ablated embryos display recovered ventricular morphology and function at 9 dpf. To clarify the discrepancy, it would be helpful if the authors could elaborate on how they assess the efficiency of ablation and the recovery of function and morphology.

*Reviewer #3:*

The authors report the consequences of heart injury in the adult zebrafish, after metronidazole mediated ablation of ventricular cardiomyocytes. Using a range of genetically modified fishes and drug treatments, they identify a hemodynamic sensitive repair mechanism, mediated by endocardial Klf2a and Notch signalling, promoting cardiomyocyte proliferation in both the ventricle and atrium, as well as atrial-to-ventricle cardiomyocyte contribution. This response reactivates a pathway involved in myocardium trabeculation, in which endocardial Notch promotes cardiomyocyte proliferation via Bmp and *erbb2* signalling.

The topic of heart repair is of high interest because of the major medical issue, but also the complexity emerging from the growing number of pathways involved in experimental models. By combining sophisticated genetics and hemodynamic measures with standard molecular and cellular analyses, the article identifies a novel repair mechanism. However, there are some important issues to address and the writing of the manuscript requires clarifications.

It is unclear why the response to injury is qualified as reprogramming. Reprogramming is used in the context of conversion of cardiac fibroblasts into cardiomyocytes, or of cardiomyocytes into pluripotent stem cells, and is associated with an erasure of epigenetic marks in the genome. What is the rationale here to refer to reprogramming and not just activation of proliferation? For example, atrial cardiomyocytes may be "pushed" from a crowded chamber to the necrotic ventricular region, where they could start to express *vmhc*. Is there a precursor state that cardiomyocytes need to revert to, to elicit a response? To which stage in normal development is this precursor state similar? What are the changes at the level of the genome? The observations that *gata4, nkx2.5* and *hand2* are activated is not convincing, since it is unclear whether this activation is in the endocardium or myocardium and whether it is required for the repair response. Hand2 is known to be activated in the endocardium downstream of Notch during trabeculation (PMID 25497097), *gata4* is known to be expressed in endocardial cells (ex PMID 16914500).

The conclusion on "regeneration" is overstated. The quantification shown as "percentage of fish" presumably corresponds to the survival rates, whereas a conclusion on regeneration would require the recovery of the tissue to be quantified. How many cardiomyocytes are lost after metronidazole treatment? How many are recovered at 72hpt, and which percentage derives from atrial cells? These quantifications are important to discuss the relative importance of the growing number of repair mechanisms. The effect of ventricular necrosis seen with *vmhc*:mCherry after 24hpt in Figure 1—figure supplement 3, Figure 3J-M, Figure 5F-H seems less dramatic compared to previous Zhang et al., 2013 publication. So how variable is the injury? Does this affect the valves, which would in itself promote retrograde flow?

The localisation of the response is always described as "throughout the heart". However, images show some degree of regionalisation, which is not discussed. The boundary of the injured area, which is also the valve region, is more affected by *klf2a*, notch reporter, *nrg1* activation. Thus, is the hemodynamic response confined to the boundary zone? Or to the valve region which is competent to activate *klf2a*? Please explain the discrepancy between the Tp1 Notch reporter, activated in the ventricle, and *notch1* expression, more striking in the atrium.

The interpretation of blebbistatin as a perturbation of hemodynamics is questionable, given that this drug also impacts proliferation (decreased in control hearts) and cell migration. Is the flow oscillatory in *gata2* mutants? Is blood flow affected in *trpv4* mutants?

The use of drugs requires control experiments to show the impact on targeted signalling pathways in this context (Figure 6).

The conclusion that endocardial Notch promotes myocardial Bmp and *erbb2* signalling is not demonstrated in this context. Whether Bmp and *erbb2* signalling are confined to, or required, in the myocardium is not shown in this context, but only based on literature in the context of trabeculation.

The Discussion section should be extended to comment the limitations/open questions of the study and the impact of the hemodynamic response for heart regeneration compared to alternative pathways, in terms of quantitative contribution, kinetics or regionalisation. The level of similarities between fish and mouse heart repair can also be discussed, rather than assumed. The last sentence is overstated.

---

## [Author Response]

[Editors' note: the authors’ plan for revisions was approved and the authors made a formal revised submission.]

Reviewer #1:In this manuscript, Chi and colleagues describe an endocardial to myocardial signalling pathway that mediates cardiomyocyte (CM) reprogramming and proliferation in the embryonic zebrafish ventricular ablation model. Increased oscillatory flow, sensed in part by the Trpv4 mechanosensory channel, is required to activate endocardial Klf2 and Notch activity, which in turn influences Bmp and Erbb2 activity in CMs. While these components have all been shown to influence cardiac development, and in many cases adult cardiac regeneration, this work provides a holistic analysis of this pathway in the embryonic setting.Overall, the manuscript is clearly written, with well presented and supported experimental results.

We thank reviewer 1 for her/his positive and thoughtful evaluation of the manuscript and her/his helpful suggestions to strengthen and deepen the conclusions of the manuscript.

Comments:1) The data for the Notch response in Figure 1—figure supplement 2 is somewhat confusing. Heightened Notch reporter activity in the ventricle is shown/discussed (panels A and B), yet Notch1b expression in panel F is clearly atrial, yet called "throughout the heart" in the text. This should be clarified.

We appreciate reviewer 1’s concern regarding the data for the Notch response to cardiac ventricle injury. Reviewer 1 is correct that *notch1b* expression in new Figure 1—figure supplement 3 (previously Figure 1—figure supplement 2) panel F and dynamic Notch reporter activity shown in new Figure 1—figure supplement 3 panel D is particularly increased in the atrium and atrioventricular canal region; however, both *notch1b* expression and Notch reporter activity is weakly present in the ventricle. These findings correlate closely to the levels of oscillatory flow after cardiac injury as shown in Figure 2. We have clarified these findings in the Result section as suggested by reviewer 1. Please see subsection “Endocardial Notch signaling controls myocardial reprogramming”.

2) In Figure 1—figure supplement 3 panel L, nkx2.5 appears to be induced in the atrium, but not ventricle, following Notch inhibition. Does this imply incomplete reprogramming of atrial CMs? Again, this should be discussed. Are these representative images?

We thank reviewer 1 for noting this issue. This data is due to incomplete inhibition of reprogramming after heat-shock induction of dnMAML Notch inhibition. Reviewer 1 is correct that new Figure 1—figure supplement 4 panel L (previously Figure 1—figure supplement 3) is meant to be a representative image. We have provided a better representative image for Figure 1—figure supplement 4 panel L, which shows more clearly the inhibition of *nkx2.5* expression and reprogramming after Notch inhibition.

3) In Figure 2—figure supplement 1, it is unclear what is being counted as a klf2/tp1 (Notch) double-positive cell, especially given the red colour of the NTR transgene as well.

We apologize for the confusion with old Figure 2—figure supplement 1, which is now revised as Figure 2—figure supplement 2 and understand the concern that ventricular cardiomyocytes and Notch reporter are expressing red fluorescence. Because Notch signaling is activated in endocardial cells after ventricular injury as we previously showed in new Figure 1—figure supplement 3 and Zhang et al., 2013, and *klf2a*:H2B-GFP is primarily activated in endothelial/endocardial cells in response to hemodynamic flow changes, we are counting endocardial cells that are co-expressing both *klf2a*:H2B-GFP and *Tp1*:nls-mCherry. To clarify these findings, we have provided better images including planar images and separation of fluorescence channels in new Figure 2—figure supplement 2 as suggested by reviewer 3. As a result, the endocardial cells can be better distinguished from the cardiomyocytes because *Tp1*:nls-mCherry appears to be more nuclear in the endocardial cells and more cytoplasmic in cardiomyocytes, and the endocardial and myocardial layers are also better separated in these planar images. In addition to these changes, we have also included new data/analyses and text to support our findings/interpretations (please see new Figure 2—figure supplement 1, subsection “Ventricular injury alters intra-cardiac hemodynamics**”** and subsection “Quantitation of Tp1/Klf2a:H2B-GFP positive cells”. However, if these new data/analyses still do not clarify our findings, we are also amenable to removing these specific data altogether as they may not be an essential part of our story/findings.

4) The authors present and support a model where alterations/increases in oscillatory flow trigger an endocardial to myocardial cascade that initiates cardiac reprogramming/proliferation. However, regeneration of the embryonic heart following full (cmlc2:NTR-based) CM ablation has been reported (Curado, 2007/2008), where presumable blood flow is minimal.

We appreciate reviewer 1’s comment. There are experimental differences between our studies and the Curado et al., 2007 that may make it more challenging to directly compare findings. For example, ablations were performed at 48 hpf in the Curado et al. studies whereas we performed our ablations later after the heart has completed cardiac looping and formed cardiac trabeculae. Furthermore, the heart does not appear to be completely ablated in the Curado et al., studies (see Figure 3, Curado et al., 2007), suggesting that the hearts are beating (presumably more weakly). Although blood flow could be weaker in the Curado ablated hearts, the actual hemodynamic flow dynamics (i.e., oscillatory flow) is unclear in these hearts since they were not examined. It is quite possible that flow is oscillatory in various regions of these *cmlc2*:NTR-based hearts, which may mediate myocardial reprogramming and regeneration. Finally, the level of recovery after *cmlc2*:NTR ablation in the Curado et al., studies is lower than the level of recovery after *vmhc*:NTR ablation. This difference in recovery could be due to weaker beating and reduced oscillatory flow in the *cmlc2*:NTR ablated hearts. To address reviewer 1’s comment, we have discussed some of these possibilities in the discussion to account for the differences in findings between studies. Please see the Discussion section.

Reviewer #2:In this manuscript, Gálvez-Santisteban et al. use a nice set of biophysical and genetic assays to characterize injury-induced cellular responses in embryonic hearts. The main conclusion is that cardiac injury alters oscillatory blood flow and induces Trpv4-mediated Klf2-Notch signaling in endocardial cells to regulate cardiomyocyte reprogramming and regeneration in erbb2 and bmp dependent manner. While this manuscript is clearly written and the data are nicely presented, a better characterization of the injury model and more robust assays for regeneration are needed before publication.

We thank reviewer 2 for her/his concerns and appreciate her/his critical evaluation of our studies. We appreciate the suggestions to clarify our data and improve our manuscript.

1) In this study, the assessment of cardiac regeneration is primarily based on pH3 staining (~4 cells per heart). More robust assays for cardiomyocyte proliferation and cardiac function will strengthen the data.

We thank reviewer 2 for her/his comment. Our assessment of cardiomyocyte proliferation by phospho-Histone H3 (pH3) staining was based on established standards as previously published (Zhang et al., 2013). However, to further validate this approach and strengthen the data, we have also performed EdU cardiomyocyte proliferation studies in our cardiac injury and regeneration model. Additionally, functional contractile and ventricular tissue recovery studies have been included to complement the cardiomyocyte proliferation experiments and show that zebrafish hearts are able to regenerate and functionally recover from cardiac injury in the *vmhc*:NTR ventricular cardiomyocyte ablation model. Please see new Figure 1—figure supplement 1 and subsection “Endocardial Notch signaling controls myocardial reprogramming”.

2) A recent study shows that the embryonic heart cannot fully recover from the ablation of a small population of cardiomyocytes. The injured hearts develop trabeculation defects which ultimately lead to heart failure. In this study, the ablation covers the entire ventricle, yet the majority of ablated embryos display recovered ventricular morphology and function at 9 dpf. To clarify the discrepancy, it would be helpful if the authors could elaborate on how they assess the efficiency of ablation and the recovery of function and morphology.

We appreciate reviewer 2’s comment and the comparison of our findings to those from the Abdul-Wajid et al., (2018). The discrepancy in findings between these studies may be due to the differences in the cardiomyocytes that are ablated. Since different cardiomyocyte ablation lines are used between the studies, we may not be ablating the same cardiomyocytes as in the Abdul-Wajid et al., studies, which are neural crest-derived and may not necessarily express *vmhc*. As a result, our *vmhc*:cherry-NTR lines may not be able to ablate these neural crest-derived cardiomyocytes, thus potentially accounting for differences in outcomes between these studies. However, we do interestingly observe in our studies that not all hearts recover from ventricular injury, suggesting the possibility that in some of our *vmhc*:cherry-NTR ablations, we are ablating these neural crest-derived cardiomyocytes. To address reviewer 2’s comment, we have included these explanations in our discussion to account for the discrepancies in findings between these studies. Please see the Discussion section.

Reviewer #3:The authors report the consequences of heart injury in the adult zebrafish, after metronidazole mediated ablation of ventricular cardiomyocytes. Using a range of genetically modified fishes and drug treatments, they identify a hemodynamic sensitive repair mechanism, mediated by endocardial Klf2a and Notch signalling, promoting cardiomyocyte proliferation in both the ventricle and atrium, as well as atrial-to-ventricle cardiomyocyte contribution. This response reactivates a pathway involved in myocardium trabeculation, in which endocardial Notch promotes cardiomyocyte proliferation via Bmp and erbb2 signalling.The topic of heart repair is of high interest because of the major medical issue, but also the complexity emerging from the growing number of pathways involved in experimental models. By combining sophisticated genetics and hemodynamic measures with standard molecular and cellular analyses, the article identifies a novel repair mechanism. However, there are some important issues to address and the writing of the manuscript requires clarifications.

We appreciate reviewer 3’s thorough review of our manuscript and her/his interest in our studies. We have carefully reviewed her/his comments and provided response and plan to address concerns.

It is unclear why the response to injury is qualified as reprogramming. Reprogramming is used in the context of conversion of cardiac fibroblasts into cardiomyocytes, or of cardiomyocytes into pluripotent stem cells, and is associated with an erasure of epigenetic marks in the genome. What is the rationale here to refer to reprogramming and not just activation of proliferation? For example, atrial cardiomyocytes may be "pushed" from a crowded chamber to the necrotic ventricular region, where they could start to express vmhc. Is there a precursor state that cardiomyocytes need to revert to, to elicit a response? To which stage in normal development is this precursor state similar? What are the changes at the level of the genome? The observations that gata4, nkx2.5 and hand2 are activated is not convincing, since it is unclear whether this activation is in the endocardium or myocardium and whether it is required for the repair response. Hand2 is known to be activated in the endocardium downstream of Notch during trabeculation (PMID 25497097), gata4 is known to be expressed in endocardial cells (ex PMID 16914500).

We appreciate reviewer 3’s query and agree that reprogramming can be used to describe the conversion of fibroblasts into another differentiated cell-type, or of a differentiated cell-type into pluripotent stem cells. However, conversion of one differentiated cell type/fate to another differentiated cell-type/fate, such as α to β cell conversion in the pancreas, is also noted to be a reprogramming process as described by several groups (PMID: 18754011, PMID: 20364121, PMID: 30760930). Thus, we have used the term reprogramming to also describe the conversion of atrial cardiomyocytes into ventricular cardiomyocyte to repair the injured ventricle as observed in our genetic-labelling lineage tracing studies (and also previously reported in Zhang et al., 2013). Furthermore, we have provided data to show cardiomyocytes may revert back to a precursor state likely through activating early cardiogenesis transcription factors as others have previously shown in cardiac regeneration studies (PMID: 23783515, PMID: 20336144, PMID: 21350179). To confirm these findings, we have used three different cardiac transcription factors involved in cardiomyocyte development rather than just relying on one. Thus, the confluence of the three transcription factor data along with our cell fate conversion/lineage tracing studies is the basis for why we use the term reprogramming in our manuscript; however, we are amenable to using other similar terms as reprogramming to describe the process including conversion and trans-differentiation when appropriate. Finally, we agree that it would be interesting to examine the extent of reprogramming of cardiomyocytes in our studies and in general, particularly from a genome/epigenome level as suggested by reviewer 3; however, such studies are beyond the scope of our current work to understand the role of hemodynamic forces during cardiac injury and repair.

The conclusion on "regeneration" is overstated. The quantification shown as "percentage of fish" presumably corresponds to the survival rates, whereas a conclusion on regeneration would require the recovery of the tissue to be quantified. How many cardiomyocytes are lost after metronidazole treatment? How many are recovered at 72hpt, and which percentage derives from atrial cells? These quantifications are important to discuss the relative importance of the growing number of repair mechanisms. The effect of ventricular necrosis seen with vmhc:mCherry after 24hpt in Figure 1—figure supplement 3, Figure 3J-M, Figure 5F-H seems less dramatic compared to previous Zhang et al., 2013 publication. So how variable is the injury? Does this affect the valves, which would in itself promote retrograde flow?

We understand reviewer 3’s concerns regarding the regeneration of the heart after ventricular injury. Our ventricular injury and regeneration studies were performed in a similar manner as we previously described in Zhang et al., 2013, which addresses issues raised by reviewer 3 including recovery of tissue, percentage of atrial cardiomyocytes that regenerate the ventricle, etc. Thus, we have cited these studies in our manuscript. Moreover, we have clarified our definition of the quantification shown as "percentage of recovered fish” in Figure 1F,1Q, Figure 4F, Figure 5Q and Figure 6K. These recovered fish represent fish that have recovered ventricular tissue and contractility at 96 hpt as previously described (Zhang et al. 2013). Please see subsection “Ventricular ablations”. Furthermore, to support these findings, we have quantified the area of the ventricular tissue at 24 hours and 96 hours post metronidazole treatment (please see subsection “Endocardial Notch signaling controls myocardial reprogramming” and new Figure 1—figure supplement 1G). To address how many cardiomyocyte are lost after metronidazole treatment, we have performed TUNEL cell death assays as we previously described in Zhang et al., 2013 and provided quantitative/statistical data of these studies to address how variable our ablations/injuries are (please see subsection “Endocardial Notch signaling controls myocardial reprogramming” and new Figure 1—figure supplement 1A-C). To complement TUNEL cell death assays, we have also provided how much ventricular tissue is lost at 24 hours post metronidazole treatment as we previously described in Zhang et al., 2013 (please see new Figure 1—figure supplement 1G). With regards to the percentage of the regenerated ventricle that derives from genetically labeled atrial cardiomyocytes, we have included this data in our manuscript (Figure 1K, Figure 4K, Figure 5U and Figure 6J). Finally, *vhmc* is specifically expressed in ventricular cardiomyocytes and thus the *vmhc*:cherry-nitroreductase line specifically ablates ventricular cardiomyocytes and not valve endocardial cells.

The localisation of the response is always described as "throughout the heart". However, images show some degree of regionalisation, which is not discussed. The boundary of the injured area, which is also the valve region, is more affected by klf2a, notch reporter, nrg1 activation. Thus, is the hemodynamic response confined to the boundary zone? Or to the valve region which is competent to activate klf2a? Please explain the discrepancy between the Tp1 Notch reporter, activated in the ventricle, and notch1 expression, more striking in the atrium.

We agree with reviewer 3 that there is some degree of regionalization of injury response. This regionalization is closely associated with hemodynamic changes during ventricular injury as we discussed in response to reviewer 1, comment 1. The discrepancy between *Tp1*:eGFP notch reporter and *notch1b* expression is because *Tp1*:eGFP does not report dynamic Notch activity due to eGFP perdurance as recently reported by Han et al., (PMID: 27357797). Thus, we have used a *Tp1:*d2GFP line to track dynamic Notch reported activity, which corresponds more closely to *notch1b* expression in injured hearts. We have clarified this point in the text (please see subsection “Endocardial Notch signaling controls myocardial reprogramming”). Additionally, we have also revised the text as suggested by reviewer 1 (please see response to reviewer 1, comment 1) as well as provided additional analyses (please see new Figure 2—figure supplement 1) and revisions in the text as suggested by reviewer 3 (please see subsection “Endocardial Notch signaling controls myocardial reprogramming”).) to address reviewer 3’s other concerns regarding regionalization of injury response.

The interpretation of blebbistatin as a perturbation of hemodynamics is questionable, given that this drug also impacts proliferation (decreased in control hearts) and cell migration. Is the flow oscillatory in gata2 mutants? Is blood flow affected in trpv4 mutants?

We agree with reviewer 3 that blebbistatin could potentially impact cell proliferation or migration. Thus, these perturbations of cardiac hemodynamics were performed using different genetic and chemical approaches to confirm findings that altering hemodynamic flow can impact the regenerative response to ventricular injury. In particular, we also used tricaine to arrest the heart by a different mechanism than blebbistatin. Additionally, we also used validated genetic approaches previously published for other cardiac hemodynamic studies (*gata2* and *trpv4* mutants; PMID: 19924233, PMID: 27221222, PMID: 25959969) to not only confirm these hemodynamic findings but also further examine potential mechanisms. Thus, to address this particular concern, we have provided more details/explanations in the text on the different approaches that were used to confirm findings as well as provide better referencing of work that has used these hemodynamic approaches before. Please see subsection “Reducing intra-cardiac hemodynamic forces impairs myocardial reprogramming”. Finally, we have provided particle image velocimetry (PIV) analyses to show that blood flow is similar between wild-type and *trpv4* -/- mutant hearts (see below Author response image 1 and Author response video 1 and Author response video 2).

**Author response image 1. respfig1:** Wild-type control and *trpv4 -/-* mutant hearts display similar blood flow patterns. High speed imaging was performed on wild-type control and *trpv4 -/-* mutant hearts at 2-3 dpf. Arrows represent particle image velocimetry (PIV) generated vectors from blood flow.

**Author response video 1. respvideo1:** *trpv4* mutant video.

**Author response video 2. respvideo2:** Wild type video.

The use of drugs requires control experiments to show the impact on targeted signalling pathways in this context (Figure 6).

We appreciate reviewer 3’s comment. We have used DMSO as a control for the drugs used in these particular studies. Furthermore, we are using specific drugs to perturb Notch, BMP and Erbb2 signaling in a similar manner as we have published and validated previously (Han et al., PMID: 27357797). Additionally, we have included an *erbb2^-/-^* mutant to confirm the AG1478 studies. Thus, we have explained that DMSO is the control in these studies and also referenced Han et al., PMID: 27357797 to clarify that the drugs used in these studies were previously verified to block respective signaling pathways. Please see subsection “Hemodynamic-mediated endocardial signaling pathways activate BMP and Erbb2 signaling to regulate myocardial reprogramming and regeneration”.

The conclusion that endocardial Notch promotes myocardial Bmp and erbb2 signalling is not demonstrated in this context. Whether Bmp and erbb2 signalling are confined to, or required, in the myocardium is not shown in this context, but only based on literature in the context of trabeculation.

We appreciate reviewer 3’s interpretation of the data in Figure 7 and agree that our Neuregulin data in Figure 7M-R is suggestive that endocardial Notch signaling during cardiac injury promotes Erbb2 signaling to regulate myocardial regeneration. However, our *BRE*:d2GFP data in Figure 7A-F does show that endocardial Notch signaling mediates myocardial Bmp signaling during ventricular injury. Thus, we have revised our text to reflect that the Neuregulin data suggests that endocardial Notch signaling may regulate Erbb2 signaling during myocardial regeneration, which has been implicated in cardiac regeneration previously (PMID: 25848746, PMID: 25830562, PMID: 27175488, please see subsection “Hemodynamic-mediated endocardial signaling pathways activate BMP and Erbb2 signaling to regulate myocardial reprogramming and regeneration). Additionally, we have not only referenced previous work that reveals that BMP signaling is present in the myocardium of zebrafish hearts but also provided better images to show that *BRE*:d2GFP Bmp reporter activity is present in cardiomyocytes of injured hearts but is altered after blocking Notch signaling (please see subsection “Hemodynamic-mediated endocardial signaling pathways activate BMP and Erbb2 signaling to regulate myocardial reprogramming and regeneration, and new Figure 7—figure supplement 1).

The discussion should be extended to comment the limitations/open questions of the study and the impact of the hemodynamic response for heart regeneration compared to alternative pathways, in terms of quantitative contribution, kinetics or regionalisation. The level of similarities between fish and mouse heart repair can also be discussed, rather than assumed. The last sentence is overstated.

We thank reviewer 3 for her/his suggestion. We have revised our discussion to include her/his suggestion. Please see Discussion section.